

# OzFlux Data: Network integration from collection to curation

Peter Isaac[1], James Cleverly[2], Ian McHugh[3], Eva van Gorsel[4], Cacilia Ewenz[5], Jason Beringer[6]

[1]OzFlux, Melbourne, VIC 3159, Australia
[2]School of Life Sciences, University of Technology, Sydney, NSW 2007, Australia
5 [3]Monash University, Melbourne, VIC 3800, Australia
[4]CSIRO Oceans and Atmospheres, Canberra, ACT 2601, Australia
[5]Airborn Research Australia, Adelaide, SA 5106, Australia
[6]School of Earth and Environment, The University of Western Australia, Crawley, WA 6009, Australia

*Correspondence to*: Peter Isaac (pisaac.ozflux@gmail.com)

10 **Abstract.** Measurement of the exchange of energy and mass between the surface and the atmospheric boundary-layer by the eddy covariance technique has undergone great change in the last two decades. Early studies of these exchanges were confined to brief field campaigns in carefully controlled conditions followed by months of data analysis. Current practice is to run tower-based eddy covariance systems continuously over several years due to the need for continuous monitoring as part of a global effort to develop local-, regional-, continental- and global-scale budgets of carbon, water and energy. 15 Efficient methods of processing the increased quantities of data are needed to maximise the time available for analysis and interpretation. Standardised methods are needed to remove differences in data processing as possible contributors to observed spatial variability. Furthermore, public availability of these datasets assists with undertaking global research efforts. The OzFlux data path has been developed (i) to provide a standard set of quality control and post-processing tools across the network, thereby facilitating inter-site integration and spatial comparisons; (ii) to increase the time available to 20 researchers for analysis and interpretation by reducing the time spent collecting and processing data; (iii) to propagate both data and metadata to the final product; and (iv) to facilitate the use of the OzFlux data by adopting a standard file format and making the data available from web-based portals. The fundamentals of the OzFlux data path include the adoption of netCDF as the underlying file format to integrate data and metadata, a suite of Python scripts to provide a standard quality control, post-processing, gap filling and partitioning environment, a portal from which data can be downloaded and an 25 OPeNDAP server offering internet access to the latest version of the OzFlux data set. Discovery of the OzFlux data set is facilitated through incorporation in FluxNet data syntheses and the publication of collection metadata via the RIF-CS format. This paper serves two purposes. The first is to describe the datasets, along with their quality control and post-processing, for the other papers of this Special Issue. The second is to provide an example of one solution to the data collection and curation challenges that are encountered by similar flux tower networks worldwide.



## 1 Introduction

Studies of the interactions between terrestrial ecosystems and the atmosphere have evolved over the last century, and the development of eddy covariance between the early studies and current networks has faced several challenges. In the mid- to late 1950s, the first eddy covariance systems were designed and built by researchers at the Australian Commonwealth

Scientific and Industrial Research Organisation (CSIRO; Dyer, 1961), when the challenge was to build the instruments with which to measure atmospheric turbulence and turbulent transport of momentum and scalars. These early experiments and those that followed made the basic measurements possible and lead to rapid advances in instrumentation technology. Even still, measurements were restricted to short, intensive field campaigns for several decades to follow. By the mid- to late 1980s, the First ISLSCP Field Experiment (FIFE) was organised to extend the studies to full growing seasons (Kim and

Verma, 1990). This intensive field campaign was a large collaborative venture that produced insight into fundamental processes of turbulence, $CO_2$ fluxes and thermodynamics of the surface layer (Kim and Verma, 1990; Norman et al., 1992; Dias and Brutsaert, 1996). Through the 1990s, eddy covariance studies were organised into similarly intensive, multi-disciplinary projects at a single location: OASIS (Observations at Several Interacting Scales) during 1994 and 1995 in New South Wales, Australia (Cleugh et al., 2004; Leuning et al., 2004) and BOREAS (Boreal Ecosystem-Atmosphere Study)

1994–1996 in Manitoba, Canada (Goulden and Crill, 1997; Goulden et al., 1997; Hogg et al., 1997; Blanken et al., 1998). These large studies were intermittent in space and time.

Shortly after the OASIS and BOREAS experiments, regional flux networks began to form: Euroflux in 1998 (Tenhunen et al., 1998), Ameriflux in 1999 (Pryor et al., 1999), and OzFlux in 2001 (Beringer et al., 2016a). The next grand challenge became to develop routines for processing the data and correcting for instrumental and physical artefacts. These algorithms

were developed independently in various laboratories and shared across the network. The network approach increased the number of investigators who could apply the eddy covariance method, which lead to rapid increase in the number of sites globally. These advances facilitated the continuous application of eddy covariance over multiple years, and this continuous data stream led to the use of flux data for process studies and model parameterisation and validation. However, data were still sparsely distributed across the globe although regional networks continued to grow, both internally and with the addition

of networks in other parts of the world (e.g., Chinaflux).

With the expansion of these flux networks, the challenge became to handle effectively the large volumes of data generated by many towers in multiple ecosystems, whilst assuring data quality and this has driven the development of several software packages for the processing of turbulence data e.g., EddyPro (Fratini and Mauder, 2014), TK3 (Maunder and Foken, 2015), EdiRe (R. Clement, University of Edinburgh, U.K.) and ECPack (van Dijk et al., 2004). At the same time, there has been a

parallel push to integrate across networks, requiring standardisation in the quality and format of data sets so as to easily combine them into larger databases like the LaThuile and FluxNet 2015 releases. This area has been advancing rapidly in recent years, but it is not as mature as instrument technology and processing algorithms that have preceded it. In OzFlux, we developed a solution (OzFluxQC) that provides a standardised set of tools and file formats to facilitate the public availability




of data in a timely manner. By offering data consistency across the network, OzFluxQC has network integration at its base and in the products that are supplied to global networks and other stakeholders, even whilst providing tools that are configurable for the specific conditions at individual sites. Most importantly, standardisation of the data format at the network level creates the possibility for data archiving that make the data discoverable, reusable and accessible for decades to come, long after the researchers who collected the data have moved on.

Provision of high-quality eddy covariance data requires that "bad" observations be removed, generating gaps in the dataset. These datasets are increasingly used for construction of local to global carbon, water or energy budgets and in modelling applications (Moffat et al., 2007), as Falge et al. (2001b) predicted. However, gaps in the data must be filled to produce defensible sums and to allow the data to be useful in modelling frameworks. Early efforts using interpolation to fill gaps were replaced by look-up tables developed from the dataset (Falge et al., 2001a; Falge et al., 2001b). In this method, gaps are filled from the measured flux during a period of similar vapour pressure deficit, solar radiation and air temperature within a ± 7-day window, which is expanded to a ± 14-day window if no other similar periods are present within ± 7 days (Reichstein et al., 2005). Gap filling via lookup tables is widely used, and the errors from filling gaps this way are smaller than for most methods, other than artificial neural networks, which produce comparably small errors when trained upon a full year (instead of the ± 7–14-day windowed approach used in the lookup tables; Moffat et al., 2007). Two types of artificial neural network (ANN) are in use for filling of gaps in flux data. The first type are feed-forward ANNs (Papale and Valentini, 2003), which are in widespread use for gap filling of flux data (Beringer et al., 2007; Moffat et al., 2007; Beringer et al., 2016b). Alternatively, a self-organising linear output (SOLO) model based upon a self-organising feature map (SOFM) is a type of ANN that is used in hydrology for its small error and resistance to overtraining (Hsu et al., 2002). SOFM has been used to evaluate the meteorological drivers of fluxes (Schmidt et al., 2011), and SOLO is effective for modelling fluxes (Abramowitz, 2005; Abramowitz et al., 2006) and gap filling of fluxes (Cleverly et al., 2013; Eamus et al., 2013) and meteorological drivers (Cleverly et al., 2016d). Because of different assumptions in various gap-filling approaches regarding the relationships between meteorological conditions and fluxes, the choice of method can affect the results obtained from flux studies and ultimately bias our understanding of biogeochemical exchange.

The eddy covariance method is used to measure the net ecosystem exchange of carbon (*NEE*). However, the expanded use of flux data globally (e.g., in biogeochemical models and in comparisons to remote sensing studies; Baldocchi, 2008) means that the components which comprise *NEE*, gross primary production (*GPP*) and ecosystem respiration (*ER*), are required by the users of flux data. There are broadly two approaches to model *GPP* or *ER* from flux data (Falge et al., 2002; Baldocchi, 2008): from nocturnal thermal sensitivity relationships (for *ER*; Reichstein et al., 2005) or light-response functions (for *GPP* and *ER*; Stoy et al., 2006; Lasslop et al., 2010). Most variations on these two approaches (e.g., $Q_{10}$ and Arrhenius thermal-response functions, and rectangular and non-rectangular hyperbola light-response functions) produce similar site ranking, conferring greater confidence in the partitioned products on the condition that the same method is used to compare fluxes across sites (Desai et al., 2008). Whereas nocturnal methods remain the most commonly applied solution to partitioning of net carbon fluxes, the combination of methods shows promise for reducing spurious correlations potentially present in these



methods in isolation (Baldocchi and Sturtevant, 2015). Some methods are systematically biased, especially in drylands and Mediterranean climates (Desai et al., 2008), in which case partitioning by multiple methods can help to avoid unrealistic results (Cleverly et al., 2013; Baldocchi and Sturtevant, 2015; Cleverly et al., 2016c). Because of the potential for bias originating from a poor choice of partitioning scheme, standardisation of partitioning methods across sites remains a

challenge, especially in OzFlux sites such as AU-Tum, where exchange with $CO_2$ stored in the canopy interacts with drainage flows (van Gorsel et al., 2007), and at AU-TTE, where standard partitioning methods generate unrealistic values of *GPP* efflux (Cleverly et al., 2016b; Cleverly et al., 2016d). For these extreme cases, we defer to Ray Leuning's oft-repeated maxim, "Know thy site," to remind us that there may not be a single, site-independent method to correctly partition carbon fluxes. At a network level, tools to employ standard approaches can be provided. In this work, we will address how the

choice of standard partitioning method affects the resultant estimates of *GPP* and *ER*.

It was recognised in the early stages of the development of OzFlux that a standard quality control (QC), post-processing, gap filling and partitioning tool was needed for the OzFlux network. The principle aims in developing such a tool were:

- To reduce site-to-site variability in the budgets of energy, water and $CO_2$ due to differences in the processing steps adopted by individual site PIs;

- To make available the cumulative wisdom of a few experienced researchers for the many OzFlux site PIs who were new to the fields of eddy covariance of surface-atmosphere interactions; and

- To provide processed data to the Australian and international research communities in a standard format and with sufficient metadata to make the data sets self-documenting.

In this paper, we will describe the production of common, standardised flux tower datasets using a processing tool developed

by OzFlux. We investigate how the choice of data processing procedures, particularly gap-filling and partitioning methods, affects estimates of *NEE*, *GPP*, *ER*, evapotranspiration (*ET*), and energy fluxes across the OzFlux network at a continental scale. Following Falge et al. (2001) and Desai et al. (2008), we anticipated that the various approaches would produce similar rankings of *GPP* and *ER* across sites i.e. that site-to-site variability would be greater than variability between methods at a single site. We further hypothesised that surface energy balance (SEB) across OzFlux will average 80%,

following trends in FluxNet (Wilson et al, 2002). Lastly, we will demonstrate how consistent, network-wide standards of data quality can be delivered to the end-user by developing a common set of procedures and making these available to the data providers (i.e., individual site PIs).

The definitions, symbols and sign conventions for carbon cycle terminology given in Chapin et al., (2006) are used throughout this paper. Refer to Table A.1. for definitions of symbols.



## 2 Materials

### 2.1 The OzFlux Tower Network

Figure 1 shows a map of Australia containing the location of the OzFlux towers with thumbnail plots of monthly average air temperature and precipitation at climate stations that are representative of the climatic regions where the bulk of towers are

located.  Only those sites for which data is available from the OzFlux Data Portal (http://data.ozflux.org.au/) are shown.  The OzFlux Data Portal contains data for 23 active sites, of which 11 receive contributions to their running costs from the Terrestrial Ecosystem Research Network (TERN), and another 13 inactive (closed) sites.  Beringer et al. (2016a) provide an overview of the OzFlux network, the location of OzFlux sites in the climate space and approximate areal extents of the ecosystems represented within the network.  Of note is the bias toward sites in native or remnant vegetation (n=24) where

carbon and water budgets are dominated by persistent ("woody") vegetation compared to the number in annually cyclic ("grassy") ecosystems (n=12).  This is important because Haverd et al. (2013a) find that two-thirds of the annual net ecosystem productivity (*NEP*) across Australia is attributable to "grassy" biomes.  The bias in site location relative to ecosystem types reflects the ad hoc nature of the network evolution over time, which is itself a feature of the network's funding history and something seen in other regional networks (Baldocchi et al., 2001).

Much of Australia receives less than 450 mm of precipitation annually, but a relatively narrow coastal band extends from the Top End (Darwin) to southern Australia (Melbourne and Hobart), and this receives much more precipitation.  There are large extremes in maximum temperature, precipitation amount and precipitation seasonality across the continent, see thumbnails in Fig. 1.  For example, northern Australia is characterised by a wet/dry tropical climate with precipitation occurring mainly during the monsoon season (November to April, see thumbnail for Darwin in Fig. 1).  Central Australia experiences a

continental climate with the largest seasonal variation in temperature (Alice Springs thumbnail, Fig. 1), and small amounts of precipitation on average, but with very large inter-annual variability (Cleverly et al., 2016a).  The southern coastal regions experience mild temperatures and a large range in the precipitation seasonality, from an autumn peak in Sydney, to little seasonality in Melbourne, and a pronounced winter peak in Perth.  The large range in precipitation amount and seasonality has resulted in a large range of biomes across Australia and OzFlux samples the majority (Beringer et al., 2016a).

OzFlux was established in 2001 as an informal network of researchers working in the field of land surface-atmosphere exchange of momentum, energy and mass (Beringer et al., 2016a).  The provision of significant funding by TERN in 2009 allowed the network to grow in size and this was done with the benefit of knowledge gained in the 8 years since the establishment of OzFlux.  It was understood from the outset that (i) resources available for operating OzFlux would be limited and (ii) that many of the site PIs would be new to the area of eddy covariance (EC) measurements of surface fluxes.

These considerations suggested that a high degree of standardisation would be required across the network to reduce the cost of establishing the new sites and to reduce the time taken to quality control and post-process the flux tower data.




## 2.2 Instrumentation Suite

OzFlux chose a standard suite of instruments but with the final decision left to site PIs dependent upon local conditions. The eddy covariance instruments at most sites consist of a CSAT3 sonic anemometer (Campbell Scientific, Logan, Utah, USA) and a Li-7500[A] open path infra-red gas analyser (IRGA, Li-cor Biosciences, Lincoln, Nebraska, USA). Three sites (AU-Tum, AU-Otw and AU-Vir) use Gill HS sonic anemometers (Gill Instruments Ltd, Lymington, UK), and one site (AU-Wrr) uses an EC155 IRGA (Campbell Scientific). Measurements of the fluxes are made outside the roughness sublayer for approximately half of the OzFlux sites using the least conservative criterion for roughness sublayer depth given in Katul et al. (1999) (RSL depth < twice canopy height). This means that care must be taken when interpreting small site to site variabilities in the fluxes over similar ecosystems because they may be due to the influence of local roughness elements at a particular site.

Logging of the eddy covariance data at most sites is done by a CR3000 or CR5000 measurement and control data logger (Campbell Scientific) with data transfer between the eddy covariance instruments and the logger by the synchronous device for measurements (SDM) protocol (Campbell Scientific). The use of this interface allows diagnostic information from the CSAT3 and the Li-7500 to be recorded and these are used during the quality control and post-processing of the data, see Sect. 3.2. The Campbell Scientific data loggers are fitted with compact flash card modules (CF100 or NL115, Campbell Scientific). Three sites (AU-Tum, AU-Otw and AU-Vir) used PC-based data logging systems built at the site PIs institution. The four components of the radiation budget are measured by CNR1 or CNR4 radiometers (Kipp and Zonen, Delft, Netherlands) and NR01 radiometers (Hukseflux, Delft, Netherlands), and most sites also measure net radiation using an NRlite (Kipp and Zonen). Additional meteorological information of air temperature and relative humidity (HMP45, HMP60, HMP155, Vaisala, Vantaa, Finland), wind speed and direction (Wind Sentry, R. M. Young, Traverse City, Michigan, USA and WindSonic4, Gill) and precipitation (various, Hydrological Services, NSW, Australia, Rimco, Campbell Scientific) are also collected. Soil measurements at all sites consist of soil temperature (TCAV, Campbell Scientific), soil water content (CS605, CS616 and CS650, Campbell Scientific) and ground heat flux (CN3, Middleton, Victoria, Australia, HFP01, Hukseflux, HFT3, Campbell Scientific). Ground heat flux plates are buried between 5 and 8 cm below the surface with soil temperature probes placed to give the average temperature in the layer between the ground heat flux plates and the surface. Soil water content sensors were buried according to manufacturer's recommendations. The typical suite of soil sensors consists of 3 ground heat flux plates and soil temperature sensors to provide some degree of spatial sampling. Soil water content measurements are made at a minimum of 2 depths, 5 and 50 cm, below the surface. Several sites, where PIs have a particular interest in soil moisture or where required by the site soil characteristics, have arrays of soil moisture sensors at multiple depths in separate soil pits to provide spatial replicates. Where the number of sensors exceeded the input capacity of the data logger, additional inputs were provided using an AM 16/32B analogue multiplexer (Campbell Scientific).





Profile systems to measure $[CO_2]$ and $[H_2O]$ in the canopy are installed at AU-Cum, AU-Tum, AU-Whr, AU-Wac and AU-Wom. Profile measurements are made at 7 heights at AU-Cum and AU-Tum and 6 heights at AU-Whr, AU-Wac and AU-Wom and all systems were built within the site PIs institution. The general sampling protocol is to draw air at a constant rate through tubing of equal length for each height being sampled and to switch these air streams through the analyser (Li-8400

or Li-7000, Li-cor) over cycles lasting for between 2 and 3.25 minutes (site dependent). Storage terms are calculated from the first and last profile measurements in each tower time step (30 or 60 minutes). The resultant storage correction terms have been applied to the AU-Tum and AU-Whr data, processing is in progress for AU-Cum and AU-Wom and will be included in future revisions of the OzFlux data set. McHugh et al. (this issue) provide details of profile systems at AU-Whr and AU-Wom.

Details of the sensor arrays at each OzFlux site are available from the OzFlux web site (http://ozflux.org.au/monitoringsites/index.html).

## 2.3 Measurement Protocols

This section addresses OzFlux sites that use Campbell Scientific data loggers. Data measurement and collection for the remaining three sites (AU-Tum, AU-Otw and AU-Vir) are described in Leuning et al. (2005). All sites fitted with Campbell

Scientific data loggers recorded the three components of the wind field, virtual air temperature and $H_2O$ and $CO_2$ concentration at 10 Hz. Slow response meteorological and soil sensors are sampled every 10 seconds, except at AU-ASM and AU-TTE, where measurement frequencies depend upon the response time for the sensor and medium (e.g., soil), including measurements at one second, 10 second, 30 second, one minute and 30 minute periods. All sites use a 30 minute averaging period, except AU-Tum, AU-Otw and AU-Vir, which use an averaging period of 60 minutes. The time stamp

refers to the end of the averaging period, which is retained throughout the OzFlux data quality control and post-processing system.

The data loggers are programmed to write the 10 Hz turbulence data, sonic diagnostic and IRGA diagnostic directly to the compact flash card. The 10 Hz data stream is also processed by the logger program to remove samples recorded when the sonic or IRGA diagnostics indicated poor data quality and this quality controlled data is then passed to the logger program's

covariance routine. On-line fluxes are calculated from these covariances at the end of each averaging period and the Webb, Pearman, Leuning (WPL) density terms are applied. Note that the data logger program does not apply a coordinate rotation to the 10 Hz turbulence data before calculating the on-line fluxes. For this reason, the on-line calculated fluxes are not used in the subsequent data processing (see Sect. 3.2) but serve as a useful instrument diagnostic during site visits. All slow response data is averaged or summed as appropriate over the 30 minute averaging period (and a 1 min. averaging period for

AU-ASM and AU-TTE; Cleverly et al., 2016d) and written to the compact flash card in separate tables for fluxes (including all covariance terms), radiation, meteorological and soil data. In addition to the 30 minute data written to the compact flash card, a subset of core data (radiation, on-line calculated fluxes, basic meteorology and soil data) is written only to the data



logger's memory and not the compact flash card, in a ring buffer of 6 months duration. This acts as a backup of core measurements in the event of a card failure.

## 2.4 Data Collection

Several methods of data collection are used across the OzFlux network. Because of the range in access to
telecommunications across the vast and sparsely populated Australian continent, a standard approach to telecommunications cannot be adopted. At all sites with Campbell Scientific data loggers, manual collection of data stored on the compact flash cards is the most basic and most robust method of data collection. However, this method is not suited to extremely remote sites and sites that are located at large distances from the site PI's institution where site visits cannot always be scheduled before the compact flash card are filled. Modems are used at all but 3 OzFlux sites to provide data in near-real time and to
provide an alternative data collection method in the event that overwrites occur on the compact flash card. The 3 sites not fitted with modems (AU-DaS, AU-Dry and AU-Wrr) are outside the mobile telephone coverage area. Two modem types are used: packet switch (ModMax, InteliMax, Maxon, NSW, Australia) and Ethernet (UniMax, Maxon). Packet switch modems are usually restricted to upload of the 30 minute average data from the sites with the 10 Hz turbulence data still read from the compact flash card, except where the site PI has disabled the LoggerNet's (Campbell Scientific) default behaviour to convert
the 10 Hz data to ASCII before transfer (AU-ASM and AU-TTE). Ethernet modems are used to upload both the 30 minute average data and the 10 Hz turbulence data. Some modems are connected via a virtual private network (Maxon) to provide a secure, static IP service to the modem.

## 2.5 Ancillary Data

OzFlux acquires additional data from the Australian Bureau of Meteorology (BoM), the European Centre for Medium Range
Weather Forecasting (ECMWF) and the Moderate Resolution Imaging Spectroradiometer (MODIS) on the TERRA and AQUA satellites. The additional data is used for gap filling flux tower data sets, as additional drivers for gap filling that contain seasonal and disturbance information and as an aid to researchers when interpreting the flux tower data. Access to the additional data is via the Australian Academic Research Network (AARNet) cloud storage facility. OzFlux maintains a shared folder on this service to which all OzFlux site PIs and invited guests have access. The additional data files are placed
in the shared storage area as they become available and users download these as required.

### 2.5.1 Automatic Weather Station data

The Australian Bureau of Meteorology (BoM) operates a network of 620 automatic weather stations (AWS) across Australia that provides observations of air temperature, humidity, wind speed, wind direction and precipitation every 30 minutes. Of the 32 OzFlux sites for which data is available, only three are farther than 50 km from an AWS and none are farther than
100 km. OzFluxQC uses data from up to three AWS for each flux tower site and from these chooses the AWS that correlates best with the tower data on a variable by variable basis. AWS data are supplied by the BoM monthly, and



processing applied by OzFlux includes checks for plausible values, linear interpolation to fill gaps shorter than two hours and 2D bi-linear interpolation to fill gaps shorter than three days. The AWS data are then output as netCDF (see Sect. 3) files to be compatible with the flux tower data format.

### 2.5.2 ACCESS-R

The second alternative source of data is the Australian Community Climate Earth System Simulator (ACCESS) numerical weather prediction (NWP) model run by the Bureau of Meteorology. The regional version of ACCESS (ACCESS-R) used for operational forecasting has a horizontal resolution of 12.5 km for the whole of Australia and a time step of one hour (Bi et al., 2013). ACCESS-R is initialised every 6 hours with an analysis field based on surface and upper air observations and between these analysis times uses information on the cloud field from polar orbiting satellites (Puri et al., 2013) to constrain

the radiation fields. OzFlux uses the 6 hourly analysis fields and the intervening 5 hours of forecast data to provide time series of radiation, meteorological and soil quantities at all of the active flux tower sites. Model output from 2011 onward is available from the BoM archive. Current ACCESS-R data is available from the BoM OPeNDAP (http://www.opendap.org/) server and is automatically retrieved by OzFlux daily. Processing of the ACCESS-R data consists of extracting cut-outs of 3 x 3 grid squares centred on the flux towers, interpolating from the ACCESS-R time step of 60 minutes to the flux tower

time step where required and calculation of instantaneous rainfall rates from the cumulative product output by the model.

### 2.5.3 ERA Interim Re-analysis

The third source of alternative data is the ERA Interim data set from the European Centre for Medium Range Weather Forecasting (Dee et al., 2011). The ERA Interim data is a re-analysis product available for the whole globe from 1979 onward. It has a horizontal resolution of approximately 75 km across Australia and is available at 6 hourly time steps with

the 00 and 12 UTC data based on observations and the 06 and 18 UTC data based upon forecast fields. The ERA Interim surface data set provides time series of radiation, meteorological and soil quantities (http://apps.ecmwf.int/datasets/data/interim-full-daily/levtype=sfc/). Processing of the ERA Interim data consists of extracting the time series data for the grid square containing the flux tower, calculating instantaneous quantities from the cumulative products and interpolating from the ERA Interim 6 hourly time step to the flux tower time step. Interpolation for

short-wave radiation data is based on the solar zenith angle and the 6 hourly rainfall data is spread evenly across all times in the interpolation period. Linear interpolation is used for all other quantities following Vuichard and Papale (2015).

### 2.5.4 MODIS

OzFlux harvests the MOD13Q1 data (Normalised Difference Vegetation Index (NDVI) and Enhance Vegetation Index (EVI)) from the TERN-AusCover Data Portal (http://www.auscover.org.au/). Cut-outs of 5 x 5 pixels around each tower are

extracted from the spatial aggregates and filtered using the MODIS quality control flags. Additional quality control includes rejecting pixels whose value is outside ± 2 standard deviations of the mean value of the 25 pixels. Values from the



remaining pixels are averaged and missing 16-day periods are filled by linear interpolation. The resulting time series of MODIS data is then smoothed using a Savitsky-Golay filter, linearly interpolated to the flux tower time step and output as netCDF files compatible with the flux tower data format. The MODIS data provide information on the state of the ecosystems around the flux towers at a temporal resolution fine enough to resolve changes due to disturbance or seasonality.

This is particularly important when filling long data gaps caused, for example, by fire because this information is often not present in the other alternate data sources.

## 3 OzFlux Data Processing

The processing system described here has several novel features. It integrates the quality control, post-processing, gap-filling and partitioning of flux tower data into a single GUI-driven package written in a simple but powerful scripting

language that is easy to maintain and extend. The system performs gap filling using meteorological, radiation and soil data from multiple alternate sources to improve the quality of the filled data for gaps longer than several days. Finally, the package encourages a deeper understanding of ecosystem processes operating at a site by presenting multiple visualisations of the data and allowing the user to quickly iterate over options at each processing level. Each step through the data processing is presented graphically to the user, who is then able to decide if the results of a particular stage are satisfactory.

This allows users and PIs to maintain a constantly updated familiarity with the conditions at their site.

The system uses a platform independent, self-documenting file format (network Common Data Format, netCDF) that allows information describing the data and the processing (metadata) to be packaged together with the data (Rew and Davis, 1990). This file format is also supported by a number of off-the-shelf tools that enable data to be easily displayed (e.g. Panoply, NASA/GISS). netCDF is the data format adopted by the Australian land-surface model CABLE (Kowalczyk et al., 2006),

and mature tools exist for publishing netCDF files via the internet (Signell et al., 2008).

The utility of the metadata contained in the netCDF files is enhanced when its possible values conform to a widely adopted schema. OzFlux has adopted the Climate and Forecasting (CF) Metadata conventions (http://cfconventions.org/, Gregory, 2003). The CF conventions define, among other things, a variable attribute called "standard_name" the value of which is chosen from a controlled vocabulary. Selecting data from a CF-compliant netCDF file using the standard_name variable

attribute avoids problems with differing variable naming schemes used by different data providers.

### 3.1 The OzFluxQC Python Suite

OzFlux has developed a suite of Python scripts, called OzFluxQC, that are integrated into a single graphical user interface (GUI) application that addresses the quality control, post-processing, gap filling and partitioning requirements associated with processing data from flux towers. The Python language was chosen because of its clear syntax, its wide spread use in

the scientific community and its encouragement of modular, easy-to-maintain and reusable program development (Oliphant,





2007). The full suite of scripts consists of over 10,000 lines of code and is available from GitHub (https://github.com/OzFlux/OzFluxQC) under the GNU Public License.

Figure 2 shows a high level diagram of the OzFluxQC data path. Processing of flux tower data is divided into 6 stages, and each stage generates a CF-compliant netCDF file. The 6 stages are identified as follows:

- L1 – level 1 processing ingests the flux tower data and writes the combined data and metadata to a netCDF file.
- L2 – level 2 processing applies a suite of quality control checks to the L1 data.
- L3 – level 3 processing applies a range of corrections to the L2 data.
- L4 – level 4 processing fills gaps in the radiation, meteorological and soil quantities.
- L5 – level 5 processing fills gaps in the flux data.

- L6 – level 6 processing partitions the gap filled NEE into GPP and ER.

All stages of processing read configuration information from text files (called control files). This convention allows site-to-site variability in the processing requirements to be abstracted from the Python code and placed in user editable files. The primary function of OzFluxQC is to post-process, gap fill and partition flux tower data, it does not process turbulence data to provide average flux values. This functionality is provided by a number of readily available software packages e.g. EddyPro

(Li-cor), TK3 (Maunder and Foken, 2015), EdiRe (R. Clement, University of Edinburgh, U.K.) and ECPack (van Dijk et al., 2004). OzFluxQC is able to use the output from these packages and skips the redundant steps at L3 when it does.

### 3.2 The Roles of OzFluxQC and DINGO

This paper describes the OzFluxQC suite of Python scripts used to quality control, post-process, gap-fill and partition data from flux towers. Beringer et al. (2016b) present a description of the Dynamic INtegrated Gap-filling and partitioning tool

for OzFlux (DINGO), a second tool that can be used for gap-filling and partitioning of flux tower data. The 2 systems are independent and serve different roles within OzFlux. The core distinction is that OzFluxQC is intended to be the operational tool used by site PIs to process their data, while DINGO is a research tool with different capabilities. The major distinctions between the two approaches can be summarised as follows.

OzFluxQC provides an integrated processing and plotting environment for all stages (L1 to L6) of flux tower data

processing, whereas DINGO performs the equivalent of L4 to L6 processing using OzFluxQC L3 data. Both systems support batch processing of data from multiple sites, and OzFluxQC provides an interactive GUI mode that allows site PIs to vary processing options to achieve the best results on a site by site, year by year basis. In keeping with its role as a production tool, OzFluxQC propagates and adds metadata throughout the processing stages and saves this with the data in netCDF files, written at the conclusion of each processing level, that are intended to be self-documenting. For example, gap

filled and partitioned fluxes at L6 contain in their variable attributes details of processing options such as the source of alternate data and settings for the neural network used during gap-filling, friction velocity and incoming shortwave radiation thresholds and other processing options usually known only to the data owner. In keeping with its role as a research tool,





DINGO provides an extensive range of diagnostic plots and outputs data in comma separated value (CSV) files for easy access.

**3.3 Data Ingestion, Quality Control and Post-processing (L1 to L3)**

The first stage of processing is designed to accept data in multiple forms, combine this data with metadata entered by the
user in the L1 control file and to write the data and metadata to a CF V1.6 compliant netCDF file. Input data can be as comma separated values (CSV) or Excel spreadsheets. OzFluxQC is designed to work with input data from two sources. For EC systems using Campbell data loggers, OzFluxQC is able to use the average covariances output by the logger. In this mode, the QC checks are done on the covariance values and the fluxes are calculated at L3 after coordinate rotation is applied to the covariances. This approach allows site PIs to quickly analyse data without having to first process the fast
turbulence data but does mean that some processing steps normally applied to the fast data (e.g. stationarity checks) cannot be performed. The second source of input data comes from software designed to process the fast turbulence data (e.g., EddyPro). With this method, the fluxes calculated by the fast data processing software are read directly into OzFluxQC at L1, the QC checks are performed on the fluxes, and the otherwise redundant corrections and flux calculations at L3 are skipped.
Quality control of surface flux data has long been recognised as an important step in producing robust surface-atmosphere data sets (Foken and Wichura, 1996, Vickers and Mahrt, 1997, Falge et al., 2001, Rebmann et al., 2005, Gockede et al. 2008, Mauder et al., 2013). Reviews of quality control steps are also given in Foken et al. (2010) and Foken et al. (2012). Quality control checks used by OzFluxQC for all variables include (i) range checks for plausible limits, (ii) spike detection, (iii) dependency on other variables and (iv) manual rejection of date ranges. Specific checks applied to the sonic and IRGA data
include rejection of points based on the sonic and IRGA diagnostic values and on either AGC or $CO_2$ and $H_2O$ signal strength, depending upon the configuration of the IRGA. The AGC and signal strengths for $CO_2$ and $H_2O$ have been found to be particularly useful for detecting periods when water droplets are present in the optical path. The spike detection routine is a modification of the standard deviation test of Vickers and Mahrt (1997), applied here to averaged data. Dependency checks allow data points of one variable to be rejected based on the value of a second variable; for example, flux
measurements can be rejected when the wind direction is outside of an acceptable range. Limits for the range check (upper and lower limits) and spike detection (the number of standard deviations about the mean) are specified for each month of the year to allow for large seasonal variability seen in some Australian ecosystems (e.g. the seasonal wet/dry Tropical savanna). L3 includes computation of meteorological variables, application of standard corrections, calculation of fluxes and optional application of profile measurements for estimating $CO_2$ storage dynamics. First, a suite of meteorological variables are
calculated and added to the data structure, including the atmospheric vapour pressure, dry and moist air densities, specific heat capacities and humidity deficits. Equations for these calculations are taken from Stull (1988) and van Dijk et al. (2004). Next, corrections are performed if they have not already been performed separately (e.g., by a SmartFlux system or by calculating the fluxes from the turbulence data), which include two-dimensional coordinate rotation (Lee et al., 2004);



corrections for frequency attenuation due to stability, sensor geometry and sensor line-averaging (Massman and Clement, 2004); re-calculation of the fluxes from the rotated, corrected covariances and meteorological variables computed in the previous step; conversion of the virtual heat flux to sensible heat flux (Schotanus et al., 1983; Campbell Scientific Inc., 2004); the WPL density terms for the effects of sensible and latent heat fluxes ($F_h$ and $F_e$, respectively) on density
measurements made by the IRGA, which is applied to correct $F_e$ and then $CO_2$ flux ($F_c$, Webb et al., 1980); correction of ground heat flux ($F_g$) for heat storage in the soil above the heat flux plates (Kustas et al., 2000); calculation of available energy as the difference between net radiation ($F_n$) and storage-corrected $F_g$ (Leuning et al., 2012); and addition of the $F_c$ storage term to the eddy covariance values of $F_c$, where applicable. The $F_c$ storage term can be supplied from values calculated externally from profile data or internally from $CO_2$ concentration at the EC instrument height.

**3.4 u\* Threshold Detection**

The underestimation of *ER* by eddy covariance measurements at night is well documented (Goulden et al., 1996; Moncrieff et al., 1996; Aubinet et al., 2000). One of the approaches used by OzFluxQC to mitigate this affect is to reject nocturnal periods when the friction velocity falls below a threshold value (although see Sect. 3.7 for an alternative partitioning method). The threshold value for u\* is determined following the change point detection (CPD) method of Barr et al. (2013).
In this approach, the time series of nocturnal *NEE* measurements is divided into windows, or "seasons", of 1000 points with successive windows overlapping by 500 points. The data for each window are divided into four equally populated temperature classes and within each temperature class the nocturnal *NEE* is averaged over 50 u\* bins. Bins with fewer than five points (30 minute averaging period) or three points (60 minute) are removed from further analysis. The u\* threshold for each temperature class in each 1000-point "season" is then found by successive fitting of a two-stage linear regression to the
binned *NEE* data while varying the centre point of the linear fit from the lowest u\* bin to the highest. The u\* thresholds for all "seasons" in a year are then averaged to get annual threshold values after rejecting those "seasons" for which the sign of the regression slopes differed from the majority (dominant mode). The fraction of "seasons" rejected for this reason is typically less than 5%.

Uncertainty in the u\* threshold value is estimated by repeating the CPD method 1000 times and randomising the time order
of the input data prior to each run (bootstrapping). Bootstrapping yields a mean threshold value and 95% confidence limits that are used to estimate uncertainty in NEE due to uncertainty in the u\* threshold. In practice, mean and confidence limits of the u\* threshold are insensitive to the number of bootstrapping iterations between 100 and 1000 runs. At some OzFlux sites, the u\*-filter can produce unrealistic results; for example, due to decoupling between the sensors deployed above a forested canopy and drainage flow beneath (AU-Tum; van Gorsel et al., 2009) or due to double-accounting of storage flows
(Aubinet, 2008) at the flat, sparsely vegetated sites in semi-arid central Australia (AU-ASM and AU-TTE; Cleverly et al., 2013). OzFluxQC offers an alternative partitioning pathway (the daytime method, Lasslop et al., 2010) for use in cases where standard methods might introduce site-specific biases.



### 3.5 Gap Filling of Drivers and Fluxes (L4 to L5)

Gap filled time series are required if site data are to be used by modellers for parameterisation and validation of land-surface models and to construct annual sums of carbon and water exchange between the land surface and the atmosphere. OzFlux data are gap-filled in two stages. Before the gaps in the fluxes are filled, gaps are filled in the environmental measurements that are used to drive gap filling of the fluxes. OzFlux sites are usually located in harsh climates that occasionally prevent access to the sites for maintenance and repair (e.g. the North Australian Tropical Transect sites in the Northern Territory). As a result, gaps in flux tower data are frequent and occasionally long. Figure 3 shows the distribution of gap occurrence (top panels) and fraction of total missing data (bottom panels) for 86 site-years across 23 OzFlux sites. The plots show results for the combined environmental drivers (left column) and the fluxes (u*, $F_h$, $F_e$ and $F_c$, right column). Short gaps of one day or shorter in duration are the most common, but the greatest contribution to the total amount of missing data comes from long gaps of 30 days or more. The high proportion of missing data due to gaps longer than 30 days means that gap filling techniques such as mean diurnal variation and marginal distribution sampling (Moffat et al., 2007) that are based on site data alone will perform poorly. As a substitute for climatology-type approaches, OzFluxQC uses data from three alternative sources to fill time series of radiation, meteorological and soil data from flux towers: one based on observations and two based on model or reanalysis outputs.

Bias is frequently observed between the data from flux towers and the alternative sources and this must be removed before the alternate data can be used for gap filling the flux tower data. Bias correction in OzFluxQC is done on 90 day windows as a compromise between providing sufficient samples for training the bias correction model and using short enough periods to avoid seasonal changes in the relationship between the flux tower and alternate data. Phase differences between the flux tower and alternate data are detected using lagged correlations and removed before performing the bias correction. Bias correction is done in each 90-day period using a linear fit calculated for each variable. For gaps longer than 90 days, the window size is increased until a minimum percentage of good data is obtained for calculating the fit. The AWS and ACCESS data sets contain more than one replicate for each variable being filled (typically three for AWS and nine for ACCESS). The replicate with the highest correlation to the flux tower data in each 90-day window is used. The effect of alternate data sources for gap filling of environmental drivers is summarised in Figures 4 (for $F_{sd}$) and 5 (for $T_a$) for 10 OzFlux sites that represent a range of ecosystems with data sets of varying gap fraction across the OzFlux network.

Quality assurance of gap-filling for environmental drives were based upon estimates of bias, variance ratio, correlation coefficient (R) and root mean square error (RMSE) in the comparison of measured *versus* modelled (ACCESS and ERAI) data sets. The variance ratio is defined here as $\sigma^2_{tower}/\sigma^2_{alternate}$ where $\sigma^2$ is the variance calculated over the 90-day window period. The variance ratio quantifies the extent to which the alternate data captures the variance in the tower data. Bias, R and RMSE values for $F_{sd}$ were comparable for both sources of alternate data, but ACCESS variance ratios were closer to unity than those for ERAI, on average (Fig. 4). The results show that both ACCESS and ERAI provide good quality data for gap filling $F_{sd}$ with ACCESS performing slightly better than ERAI. The results for $F_{su}$, $F_{ld}$, $F_{lu}$, $F_n$ and $F_a$





(not shown) are similar. Figure 5 shows the performance of alternate data sources (AWS, ACCESS and ERAI) on gap filling of air temperature. Bias in air temperature was small at most sites and variance ratios were close to unity. Both R and RMSE show that the AWS data for air temperature was a significantly better match (higher R, lower RMSE) to the tower data than ACCESS. ERAI data performs poorly compared to the AWS and ACCESS data. The results of comparisons for all radiation, meteorological and soil quantities (not shown) confirm that the combination of NWP model or re-analysis data for gap filling radiation and soil variables and AWS data for gap filling meteorological variables provides the best overall performance.

OzFluxQC uses the SOLO ANN (Hsu et al, 2002) to gap fill the fluxes. Details of the SOLO neural network design and operation are given in and Abramowitz (2005) and its use in Australian conditions is discussed in Cleverly et al (2013) and Eamus et al (2013). In SOLO, SOFM performs the ANN equivalent of a principal component analysis to determine the relationships amongst environmental drivers. Next, SOLO performs the ANN equivalent of a multiple linear regression between the results of SOFM and the fluxes. The choice of fluxes and environmental drivers is configurable by the user, but the default settings are to fill u* (with $Ws$, $F_n$, $T_a$ and $q$ as drivers), $F_h$ ($F_a$, $T_a$ and $Ws$), $F_e$ ($F_a$, $q$, $T_a$ and $Ws$) and $F_c$ ($F_n$, $F_g$, $q$, $VPD$, $T_a$ and $T_s$). A u* filter was applied to the $F_c$ data prior to gap filling, and annual thresholds were determined as described in Sect. 3.5. The proportion of nocturnal $NEE$ measurements remaining after application of the u*-filter varied between 11% and 23% across all sites in the OzFlux network. The choice of window size for gap filling the fluxes is a compromise between a long enough period to provide sufficient points for training the ANN and short enough to allow the ANN to accommodate seasonal changes in the relationships between the drivers and the fluxes. The default window size is 60 days, but the window size and all other ANN parameters can be set by the user. Multiple interactive runs are generally required to find the best combination of window length and ANN parameters for any given site. All statistics were improved (smaller RMSE, higher R and variance ratio closer to 1) when gap-filling was performed on windowed data rather than on entire time series, see Figure 6. RMSE was least sensitive to window length, but shorter window lengths were associated with values of R that were higher and values of the variance ratio closer to unity, much in the same way that windowed lookup tables perform with small errors when the gaps are small enough (Moffat et al., 2007). However, care must be taken when gap lengths approach the window size. OzFluxQC deals with this by detecting when the window does not contain enough points and increases the window size until the minimum points criteria is met.

### 3.6 Partitioning of NEE (L6)

A common goal of flux tower research is to partition observed $NEE$ into the component fluxes of $GPP$ and $ER$. This is an ill-posed problem where we infer two large numbers from their small difference (Baldocchi, 2008). OzFluxQC offers three methods for estimating $ER$ from flux tower data: two are based on the nocturnal thermal-response approaches (Papale et al., 2006), and one is based on day time light-response functions (Lasslop et al., 2010). The two nocturnal approaches apply a u*-filter to nocturnal measurements of $F_c$, but they apply different models to estimate ER from the filtered data. The first nocturnal approach applies a modified Arrhenius equation (Lloyd and Taylor, 1994):





$$ER = ER_{10} \exp \left( E_0 \left( \frac{1}{T_{ref} - T_0} - \frac{1}{T - T_0} \right) \right) \tag{1}$$

In this equation, $T_{ref} = 10$ C is the reference temperature, $T_0 = -46.04$ C is a model parameter chosen by Lloyd and Taylor (1994) to fit their soil respiration data, $ER_{10}$ is the base respiration rate at the reference temperature and $E_0$ is the activation energy. Either air temperature or soil temperature can be used in the implementation of Eqn. 1 in OzFluxQC (Lasslop et al.,

2012). The values of $ER_{10}$ and $E_0$ were estimated from nocturnal, u*-filtered measurements of $F_c$ using the Levenberg-Marquardt damped least squares method. $E_0$ values were determined using a window size of one calendar year, and $ER_{10}$ was estimated using a window size of 15 days and an overlap of 10 days. The resulting values for $ER_{10}$ and $E_0$ were quality controlled as described in Lasslop et al. (2010), and missing parameter values were filled using linear interpolation. $ER$ was then estimated using Eqn. 1 with the gap filled air or soil temperature, as appropriate.

The second nocturnal approach also used nocturnal, u*-filtered $F_c$ measurements, but here respiration was modelled using SOLO, with $T_a$, $T_s$, soil water content and, optionally, MODIS EVI data as drivers. Due to the small amount of data retained after the u*-filter was applied, we use window lengths of a minimum of one year in length for this approach. Inclusion of MODIS EVI as a driver improved the fit of the SOLO output when long window lengths were used. The ANN was trained only on observations that have passed all quality control checks; that is, gap filled data were not used. After training, SOLO

was used to produce predictions of $ER$ at all time steps in the tower record (i.e., based upon gap-filled fluxes).

The third approach for estimating $ER$ used daytime data, with a rectangular hyperbola fit between $F_{sd}$ and $NEE$ (following Lasslop et al., 2010):

$$NEE = \frac{\alpha \beta F_{sd}}{\alpha F_{sd} + \beta} + ER_{10} \exp \left( E_0 \left( \frac{1}{T_{ref} - T_0} - \frac{1}{T - T_0} \right) \right) \tag{2}$$

The second term on the right side of Eqn. 2 is nocturnal $ER$ (i.e., in the absence of light; Eq. 1). In the first term, $F_{sd}$ is

incoming shortwave radiation, $\alpha$ is the initial slope of the light response curve and $\beta$ is the photosynthetic capacity at light saturation, which is modified from its maximum value $\beta_0$ by an exponential function of vapour pressure deficit ($VPD$):

$$\beta = \begin{cases} \beta_0 \exp \left( -k \left( VPD - VPD_0 \right) \right), VPD > VPD_0 \\ \beta_0 \qquad\qquad\qquad\qquad\quad, VPD \leq VPD_0 \end{cases} \tag{3}$$

In this model, $T_{ref}$, $T_0$ and $VPD_0 = 1$ kPa all have fixed values. $E_0$ is calculated for each calendar year as described above. $\alpha$, $\beta$, $ER_{10}$ and $k$ are all found by non-linear curve fitting using the Levenberg-Marquardt method for 15 day windows with 10

days overlap between consecutive windows. Quality control of the estimated parameters follows Lasslop et al (2010). With all of the above methods, the final $ER$ time series was constructed from observations of $F_c$ at night ($F_{sd} < 10$ W m$^{-2}$) when u* is above the threshold and modelled $ER$ at all other times. The final $NEE$ time series was constructed from gap-filled $F_c$ during the day ($F_{sd} >= 10$ W m$^{-2}$) and $ER$ at night. GPP was then calculated from the $NEE$ and $ER$ time series as:



$$GPP = -NEE + ER \qquad\qquad (4)$$

## 4 OzFlux Data Access

There is an increasing awareness within the research community and those who fund research that data should be easily available and re-usable. Making data re-usable means that it must be accompanied by sufficient information, or metadata, to
allow the user to decide if the data is fit for their purpose. Within the FluxNet community, the BADM templates are one approach to collecting this metadata. In OzFlux, metadata is provided by OzFluxQC and bundled directly with the data in the netCDF file. Alternative file formats can be made available to facilitate collaboration with other networks (e.g., FluxNet) or research organisations (e.g., NASA's SMAP project; http://smap.jpl.nasa.gov/). In these latter cases, where data is provided as CSV files, OzFluxQC can include metadata in the file header, as negotiated on a case-by-case basis.
OzFlux contributed 17 site-years of L3 data from 4 sites to the LaThuile FluxNet data set. At the April 2016 update of the FluxNet2015 synthesis, there were 62 site-years of L3 data from 15 OzFlux sites and this will be extended to 86 site-years from 23 sites by the July 2016 update. The OzFlux submission to FluxNet is available from the FluxNet2015 download page (http://fluxnet.fluxdata.org/data/fluxnet2015-dataset/) and from the FluxNet collection on the ODP.

### 4.1 The OzFlux Data Portal

The primary repository for OzFlux data is the OzFlux Data Portal (ODP, http://data.ozflux.org.au/). Data on the ODP are organised into collections such that each collection represents a single flux tower. The collections can be viewed as a list or via a map interface. The collection home pages contain a brief description of the site and the temporal coverage of the collection. Citation information is also provided including a persistent identifier (handle) that resolves to the collection when typed into the address bar of a web browser (hdl.handle.net). The handle simplifies the citation of data sets and increases the
visibility of the data. Like all TERN datasets, the OzFlux collections can be browsed by the public without login credentials. OzFlux L3 data is stored on the portal in annual files, although some PIs also choose to provide L4 to L6 data as well. The metadata associated with each netCDF file (the global and variable attributes) can be viewed from the collection web page. This enables users to assess the suitability of the data without first downloading the netCDF files. The collection page entry for each file also indicates when access to the data is restricted. Site PIs may choose to restrict access to data for up to 18
months when it is necessary to protect student intellectual property, although a waiver for data usage that does not conflict with the student's research can be negotiated with the site PI. The download of files that are not restricted does not require the user to have an account on the portal or to be logged in. More than 90% of the data on the ODP is unrestricted.

The default license applied to data uploaded to the ODP is the TERN BY-SA-NC license. This is a variant of the Creative Commons V3 license that requires attribution of the data owner when the data is used (BY), states that any derivative
products are also covered by the same license (share alike or SA) and allows commercial use only if authorised by the data owner (NC). TERN reserves the right to change the licence at their discretion. Collection metadata is made available by the



ODP in Registry Interchange Format – Collections and Services schema on a publicly available server. Organisations such as Research Data Australia (https://researchdata.ands.org.au/) and the TERN Data Discovery Portal (http://portal.tern.org.au/) can then harvest this metadata, making the ODP collections visible via their own web presence. Source code for the ODP is publicly available (https://code.google.com/archive/p/eddy/).

## 4.2 The OzFlux OPeNDAP Server

The Open-source Project for a Network Data Access Protocol (OPeNDAP, http://opendap.org/) provides a mechanism for serving data via the internet. Access to an OPeNDAP server can be via a web browser or via OPeNDAP-aware programmes such as Panoply (http://www.giss.nasa.gov/tools/panoply/). A major feature of OPeNDAP is that the server can provide subsets of the data, as configured by the users, allowing them to view and manipulate the data without downloading the complete file. OzFlux has implemented an OPeNDAP server (http://dap.ozflux.org.au/), and unrestricted data from the OzFlux data set is available from this server. The data on the OPeNDAP server is stored by site and then by processing level (L3 to L6). Within each processing level there are "site_pi" and "default" data sets. The "default" data sets are processed by OzFlux using the standard methods described in this paper, using the default options. This achieves a high level of standardisation for the processing across all sites but may not produce the best results at an individual site. The "site_pi" data sets are processed by the site PIs and may use non-standard methods or options based on the PIs knowledge of their own site. Data available from the OzFlux OPeNDAP server is combined into single, multiple-year files for each processing level at each site.

## 5 Results

### 5.1 Surface energy balance closure

The closure of the surface energy budget (SEB) is an important diagnostic for eddy covariance measurements (Aubinet et al. 2000; Leuning et al., 2012; Stoy et al., 2013). In their analysis of data from the LaThuile FluxNet data set, Stoy et al. (2013) found an overall value for the SEB closure of $0.84 \pm 0.2$ and concluded that, among other factors, non-closure of the SEB is related to landscape scale heterogeneity. In an analysis of the same data set but using a slightly different methodology, Leuning et al. (2012) found a median value for SEB closure of 0.75 for half-hourly data, rising to 0.90 when daily averages were used. We investigated the SEB closure for 86 site-years of data across 23 OzFlux sites following the approach of Leuning et al. (2012). Daily values of $(F_h+F_e)$ and $(F_n-F_g)$ were calculated for each site using non-gap filled data on days when more than 80% of records were present. Figure 7 shows a histogram of SEB ratios ($[F_h+F_e] / [F_n-F_g]$) for the OzFlux sites. 50% of site-years had an SEB ratio of 0.89 or higher, and 80% had an SEB ratio greater than 0.80. Only 8% of site-years have an SEB ratio greater than 1.00. These values for SEB closure are similar to those found in recent reviews (Stoy et al., 2013; Leuning et al., 2012).



## 5.2 Comparison with EddyPro

Fluxes calculated by OzFluxQC from the 30 minute covariances may differ from those calculated from the 10 or 20 Hz turbulence data by packages such as EddyPro and EdiRE due to differences in the processing algorithms (Mauder et al., 2008, Fratini and Mauder, 2014). The quality control tests applied to the turbulence data during processing are also different
to those applied when the logger calculates the covariances online and this may also result in further differences between the values from the two approaches. To investigate this, we compared fluxes calculated by OzFluxQC from the 30 minute covariances with those calculated from the 10 Hz turbulence data using EddyPro for a two-month period at the AU-How site. The EddyPro processing options were set to mimic those used by OzFluxQC (no spike removal on 10 Hz data, block averaging, 2D coordinate rotation, Massman frequency corrections). The results of the comparison are shown in Figure 8,
along with the regression statistics (slope, offset and correlation coefficient). There was no evidence of bias or scatter between the two sources of flux calculations, confirming that calculating fluxes from the 30 minute covariances with OzFluxQC produces equivalent values to those calculated from the turbulence data using EddyPro with equivalent processing options.

## 5.3 Comparison of annual sums

We compare annual sums of *NEE*, *GPP* and *ER* estimated using 5 different methods at 9 OzFlux sites for the calendar year 2013. The methods used were: OzFluxQC nocturnal method with the Lloyd-Taylor respiration model; REddyProc nocturnal method with the Lloyd-Taylor respiration model; OzFluxQC nocturnal method with an ANN respiration model; DINGO nocturnal method with an ANN respiration model and the OzFluxQC day time method. Annual sums from the FluxNet nocturnal method with the Lloyd-Taylor respiration model and the FluxNet day time method were calculated for 6 sites
(FluxNet data is not yet available for the other 3 sites). 2013 was chosen based on the overlap of years available from the OzFlux and FluxNet 2015 data sets and to minimise the extent of gap filling in the tower data.

The general rankings of sites were consistent across methods, with some notable exceptions, see Figure 9. Daytime approaches by OzFlux, REddyProc and FluxNet produced 20–57% (median 35%) smaller values of *ER* at all sites compared to the nocturnal methods. The under-estimation of *ER* by the daytime methods was particularly noticeable at AU-Cum, AU-
Tum and AU-Whr. By contrast, nocturnal, u*-filtered approaches using neural networks to estimate *ER* (OzFlux and DINGO) produced very similar results at all sites (median 7% difference, range 0 to 14%). The largest difference (14%) occurred for AU-Tum, which was likely to be due to differences in the application of the nocturnal u*-filter (DINGO used only the first three hours after sunset, see Beringer et al, this issue). The nocturnal, u*-filtered approaches using the Lloyd-Taylor *ER* model shows the largest variability between methods within a single site (median 24%, range -6 to 170%),
particularly at AU-Cum, AU-DaS and AU-Tum. The within-site variability can be explained by differences in the approach and data used by each method. For example, OzFluxQC determines a value of $E_0$ annually whereas REddyProc uses a single value for all years (details for FluxNet not available at the current time). OzFluxQC and FluxNet calculate the u* threshold





using the CPD method but REddyProc uses the method described in Papale et al (2006). The OzFluxQC CPD method gives a u* threshold for AU-Tum of 0.58 m s$^{-1}$ compared to values of 0.24 m s$^{-1}$ for the same method implemented by FluxNet and 0.28 m s$^{-1}$ for the Reichstein method used in REddyProc. To examine the effect of these different u* threshold values on the results we repeated the AU-Tum processing with OzFluxQC and REddyProc using the u* threshold applied in the FluxNet

processing. Annual sums for the reprocessed data are given in Table 1. The median difference between the methods reduces to 12%, 4% and 22% for *NEE*, *GPP* and *ER* respectively. This is a substantial reduction but disparities remain. Small differences in results from different methods are to be expected and may be another indicator of uncertainty if, *a priori*, we have no basis for deciding which method is correct at a given site or under particular circumstances. However, different implementations of the same algorithm (e.g. the nocturnal method with the Lloyd-Taylor respiration model) are expected to

agree and where differences exist, these need to be resolved.

## 6 Conclusions

In the 36 years since Webb et al (1980), there has been a concerted effort to standardise algorithms for calculating and correcting fluxes. As noted 15 years ago by Falge et al (2001), there has not been the same advance in standard approaches to gap filling and partitioning, although progress has been made (Reichstein et al, 2005; Papale et al 2006; Moffat et al,

2007). As well as a standard approach to gap filling and partitioning, the advance from individual site programs to global syntheses requires a corresponding advance in the standardisation of data archiving, data visibility and data accessibility to promote the re-use of flux network data, especially by those outside the eddy covariance community. The FluxNet LaThuile and 2015 synthesis data sets and the FluxNet BADM initiative are valuable moves in this direction.

In this paper we have described an integrated suite of Python scripts, OzFluxQC, that provides a powerful and flexible tool

for processing flux tower data that addresses some of these issues. A key feature of this processing path is the use of netCDF files that provide a cross-platform file format, that allows metadata to be transported with the data in a self-describing file format and that is widely supported in atmospheric and oceanic research communities.

The primary design goals for OzFluxQC were (i) to provide a standard processing method that makes the expert knowledge of a few available to a large audience of users with various levels of expertise and (ii) to reduce the time required to produce

gap filled and partitioned fluxes so that PIs have more time to concentrate on the science. Novel aspects of the OzFluxQC suite include the integration of all steps into a single framework, interactive processing via a GUI or batch processing of multiple sites, the use of netCDF as the file format so that metadata travels with the data, the use of AWS and NWP models as sources for alternate data when gap filling meteorological data from towers, the implementation of several partitioning methods in a single package and the integration of data processing and data visibility and availability made possible by the

use of the netCDF format.

We have demonstrated the utility of this package using a data set of ~86 site-years from ~23 OzFlux sites. Fluxes calculated with this package show excellent agreement with those calculated from 10 Hz turbulence data. Overall, the SEB closure across the OzFlux network is similar to that found for other flux tower synthesis data sets.



While gaps of short duration make up the majority of gap occurrences in the OzFlux data set, gaps longer than 5 days account for most of the missing data and gaps longer than 30 days are a significant component. To improve the filling of long gaps we use data from AWS, NWP models and global re-analysis. For radiation quantities we find that the global re-analysis at ~75 km horizontal resolution performs only slightly worse than the NWP output at 12.5 km resolution and that

both are highly correlated to flux tower data. For meteorological quantities we find that AWS data performs significantly better than NWP output with re-analysis data performing worst of the alternate sources. These results demonstrate the utility of NWP output and global re-analysis products for gap filling radiation but suggest local AWS data is a better choice for meteorological quantities.

When gap filling fluxes, the user often has a choice of the window size over which the gap filling method is to be trained.

For the ANN used in OzFluxQC, we find the best performance with a window size of 60 days and that the performance degrades with increasing window length. Applying the ANN to the whole data set gives the worst performance. The most likely explanation for this behaviour is that the use of short window lengths allows the ANN to re-train and take in to account seasonal changes in the relationship between the drivers and the target flux. It is possible that using more complex ANN designs and training these harder would improve the statistics for long windows but this carries an increased risk of an

over-trained network and consequent introduction of spurious predictions.

Finally, we have used results from 9 OzFlux sites to compare annual sums of *NEE*, *GPP* and *ER* for 2013 calculated using 7 different methods (3 from OzFluxQC, 2 from FluxNet 2015, 1 each from REddyProc and DINGO). We find that day time partitioning methods using a combination of light-response curve for *GPP* and an Arrhenius-style equation for *ER* under-estimate *ER* and over-estimate *NEE* at most sites. Site to site variability follows the same trends across all partitioning

methods. Variability between different implementations of the same method at a single site is significant though still smaller than the site-to-site differences.

## 7 Future Directions

The development of OzFluxQC has provided OzFlux with a powerful tool for quality control, post-processing, gap filling and partitioning of flux tower data that is now in routine use across the network. With this base established, several areas

present themselves as potential future directions for the OzFlux data set and OzFluxQC.

One of the most important areas for future work is to harmonise the OzFlux data with FluxNet. The OzFlux netCDF files are a rich source of metadata but need to be supplemented by the completion of the FluxNet BADM templates for all OzFlux sites. This will make local knowledge about the sites available to the wider research community and aid in the use and interpretation of the OzFlux data. Other areas for rationalising the approaches across different networks are variable names

and the terminology for processing levels.

A major area of future work is resolving the disparities in annual sums of *NEE*, *GPP* and *ER* given by different implementations of the same processing method, as shown in Sect. 5.3. A related area of research is resolving the disparities



between different methods, for example day time approaches under-estimate *ER* and over-estimate *NEE* at OzFlux sites compared to night time, u*-filter approaches.

Characterisation of the uncertainty in the carbon and water budgets from flux towers is an important area for future work. We propose to estimate the uncertainty due to random error in the eddy covariance measurements using the paired-observation technique of Hollinger and Richardson (2005). Uncertainty due to the distribution of u* threshold and gaps in the data will be estimated from multiple runs by selecting threshold values from the distribution, selecting a gap scenario and then repeating the gap filling and partitioning process to construct distributions of *NEE*, *GPP* and *ER*. This will require a large number of runs, of the order of $10^4$, to cover the range of threshold values and gap scenarios, and will be computationally expensive making this feature better suited to a stand-alone utility, rather than as part of the interactive OzFluxQC.

Footprint visualisation is an important aid to interpretation of flux tower data, especially at inhomogeneous sites. We plan to integrate existing code that compiles a footprint climatology using the Kormann and Meixner (Kormann and Meixner, 2001) and Kljun 2D (Kljun et al, 2015) models into OzFluxQC over the next 6 months.

### Acknowledgements

The authors would like to thank Drs Ray Leuning and Helen Cleugh (CSIRO) for the leading role they played in establishing OzFlux. We also thank all members of the OzFlux community, whose efforts have been critical to the success of the OzFlux network, and the Terrestrial Ecosystem Research Network for its crucial support of OzFlux. Finally, we also wish to thank Assoc. Prof. Kuo-lin Hsu (University of California, Irvine) for providing the source code for the SOLO artificial neural network. Beringer is funded under an ARC FT (FT1110602).

The ODP was developed by Monash University eResearch Centre and funded by the Australian National Data Service as part of the Monash ARDC-EIF Data Capture and Metadata Store project.



## Appendix 1: OzFlux Variable Names

Table A.1 presents a list of the symbols and names used for the main variables in the OzFlux tower data.

| Symbol | Name | Symbol | Name |
|---|---|---|---|
| $Ah$ | Absolute humidity, g m$^{-3}$ | $GPP$ | Gross primary productivity, µmol m$^{-2}$ s$^{-1}$ |
| $Cc$ | CO2 concentration, µmol mol$^{-1}$ | $NEE$ | Net ecosystem exchange, µmol m$^{-2}$ s$^{-1}$ |
| $ER$ | Ecosystem respiration, µmol m$^{-2}$ s$^{-1}$ | $Precip$ | Total precipitation, mm |
| $F_a$ | Available energy, W m$^{-2}$ | $ps$ | Surface pressure, kPa |
| $F_c$ | CO2 flux, µmol m$^{-2}$ s$^{-1}$ | $q$ | Specific humidity, kg kg$^{-1}$ |
| $F_e$ | Latent heat flux, W m$^{-2}$ | $RH$ | Relative humidity, % |
| $F_g$ | Ground heat flux, W m$^{-2}$ | $SHD$ | Specific humidity deficit, kg kg$^{-1}$ |
| $F_h$ | Sensible heat flux, W m$^{-2}$ | $Sws$ | Soil water content, m m$^{-1}$ |
| $F_{ld}$ | Downwelling long-wave radiation, W m$^{-2}$ | $T_a$ | Air temperature, C |
| $F_{lu}$ | Upwelling long-wave radiation, W m$^{-2}$ | $T_s$ | Soil temperature, C |
| $F_m$ | Momentum flux, kg m s$^{-2}$ | $ustar$ | Friction velocity, m s$^{-1}$ |
| $F_n$ | Net radiation, W m$^{-2}$ | $VPD$ | Vapour pressure deficit, kPa |
| $F_{sd}$ | Downwelling shortwave radiation, W m$^{-2}$ | $Ws$ | Wind speed, m s$^{-1}$ |
| $F_{su}$ | Upwelling shortwave radiation, W m$^{-2}$ | $Wd$ | Wind direction, deg |

For $ER$, $NEE$ and $GPP$ we use the units µmol m$^{-2}$ s$^{-1}$ for 30- or 60-minute values. Daily, monthly and annual totals are given in units gC m$^{-2}$ day$^{-1}$, gC m$^{-2}$ month$^{-1}$ and gC m$^{-2}$ year$^{-1}$ respectively.





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





| | *NEE* (gC m$^{-2}$ year$^{-1}$) | *GPP* (gC m$^{-2}$ year$^{-1}$) | *ER* (gC m$^{-2}$ year$^{-1}$) |
|---|---|---|---|
| OF(NT/LT) | -1243 | 2762 | 1519 |
| FN(NT/LT) | -1401 | 2683 | 1235 |
| REP(NT/LT) | -1244 | 2795 | 1551 |

Table 1: Annual sums of *NEE*, *GPP* and *ER* for 2013 at AU-Tum processed with the u* threshold value set to 0.24 m s$^{-1}$.





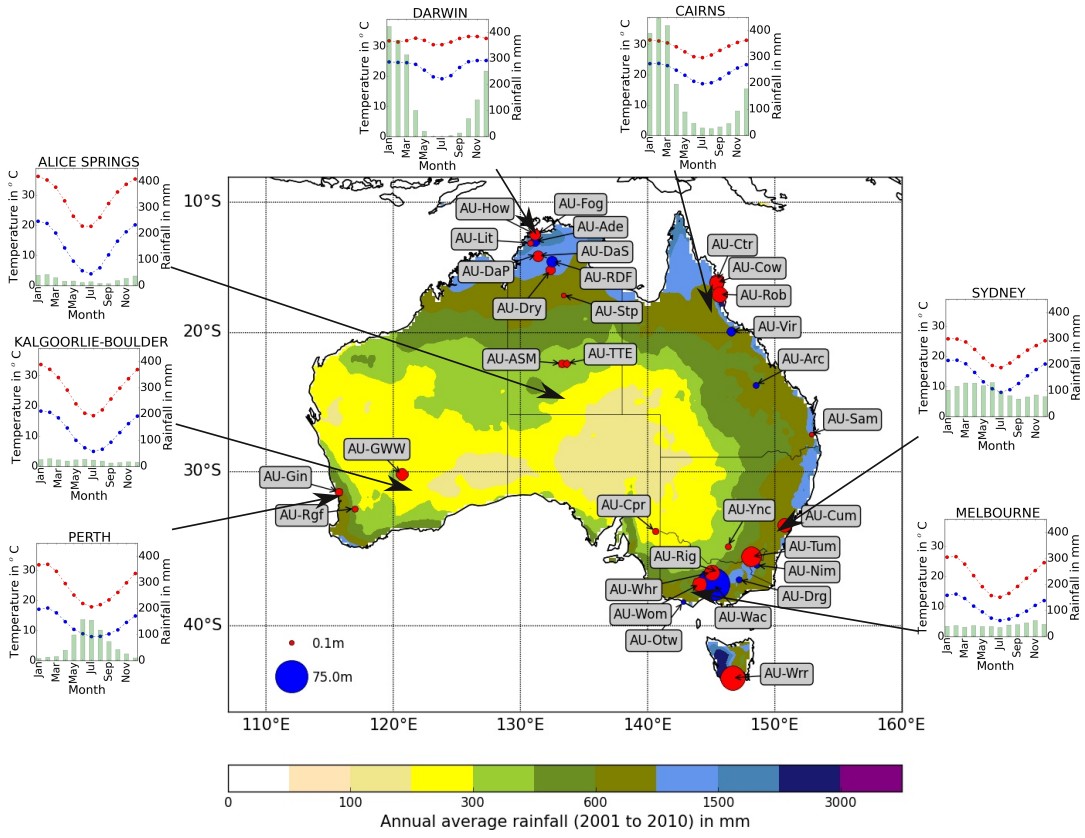

**Figure 1: Map of Australia showing the location of OzFlux sites (red for active sites, blue for inactive) and thumbnail plots of monthly average air temperature and precipitation at Bureau of Meteorology sites representative of the tower locations. The diameter of the symbol marking the tower locations is proportional to the canopy height, see scale at bottom left.**





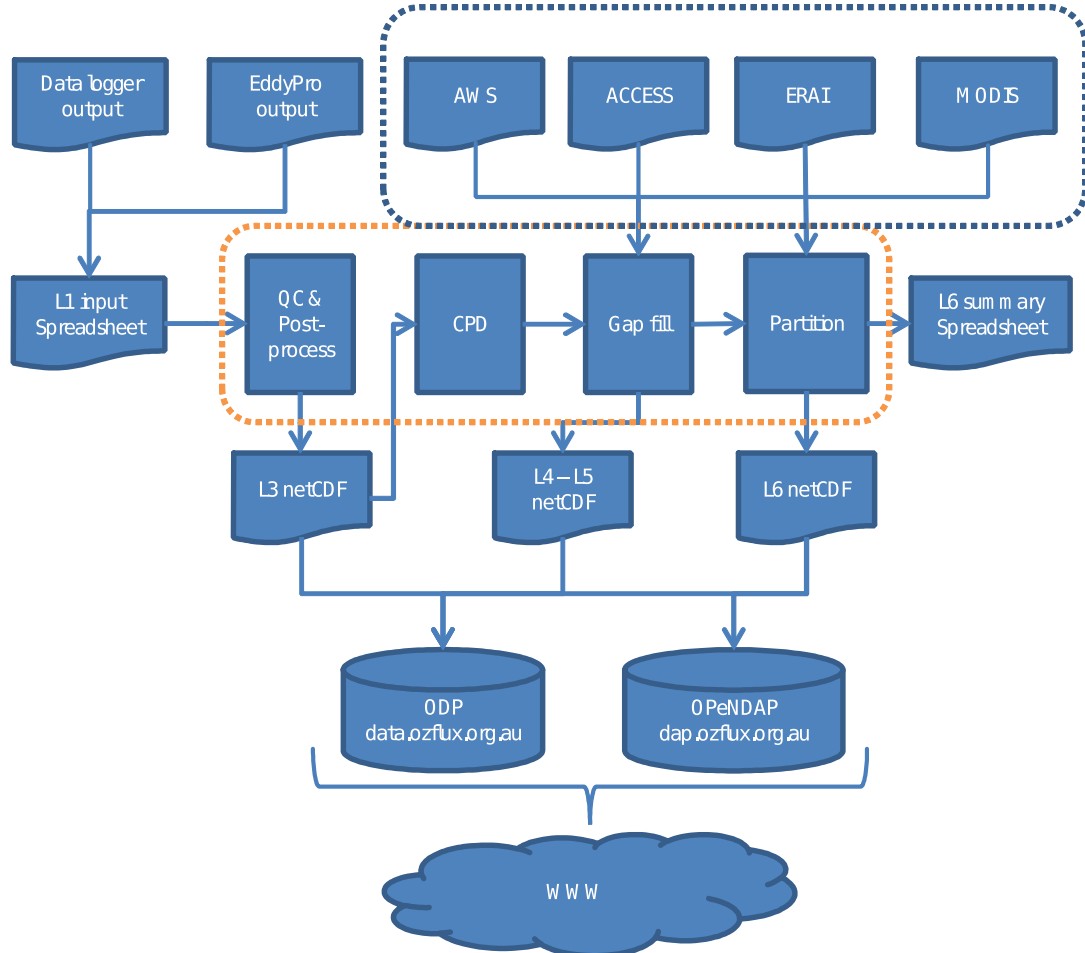

**Figure 2: High level diagram of the OzFlux data path. Data input can be the averaged covariances output by the data logger ("Data logger output") or fluxes calculated by EddyPro or similar from the fast turbulence data ("EddyPro output"). Processes in the dotted orange line are contained in OzFluxQC, processes in the dotted blue line are utilities provided with OzFluxQC.**




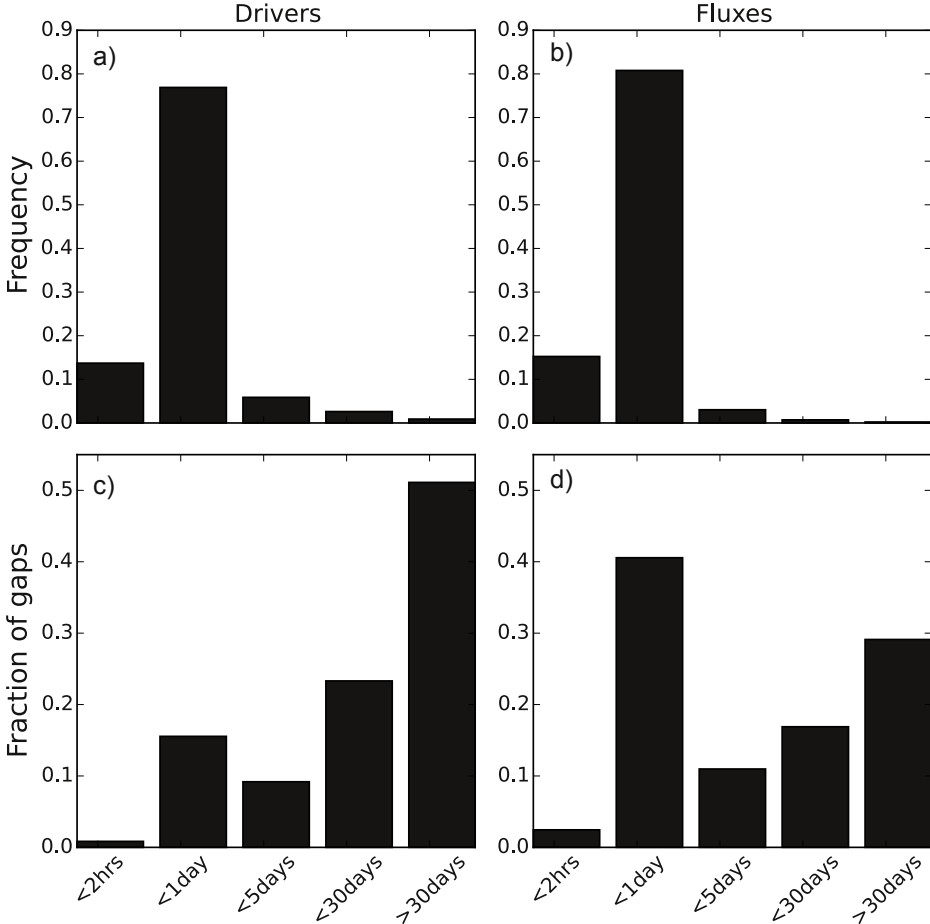

**Figure 3: Histograms of gap frequency by duration for a) drivers and b) fluxes and histograms of the fraction of total gap length for c) drivers and d) fluxes.**




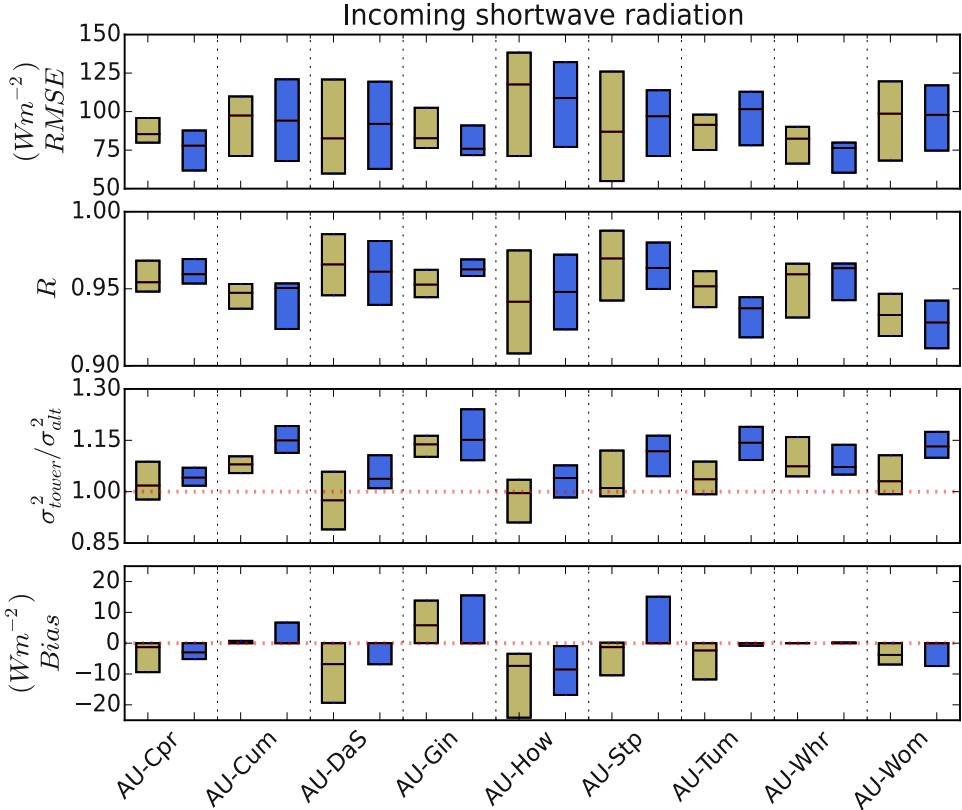

**Figure 4: Box plot of bias, variance ratio, correlation coefficient and root mean square error for the comparison of tower and alternate incoming short wave radiation at 9 OzFlux sites. Statistics for ACCESS-R are plotted in green and ERA Interim in blue. Box lengths represent the 75% and 25% quartiles and the median is plotted as a solid line across the boxes.**





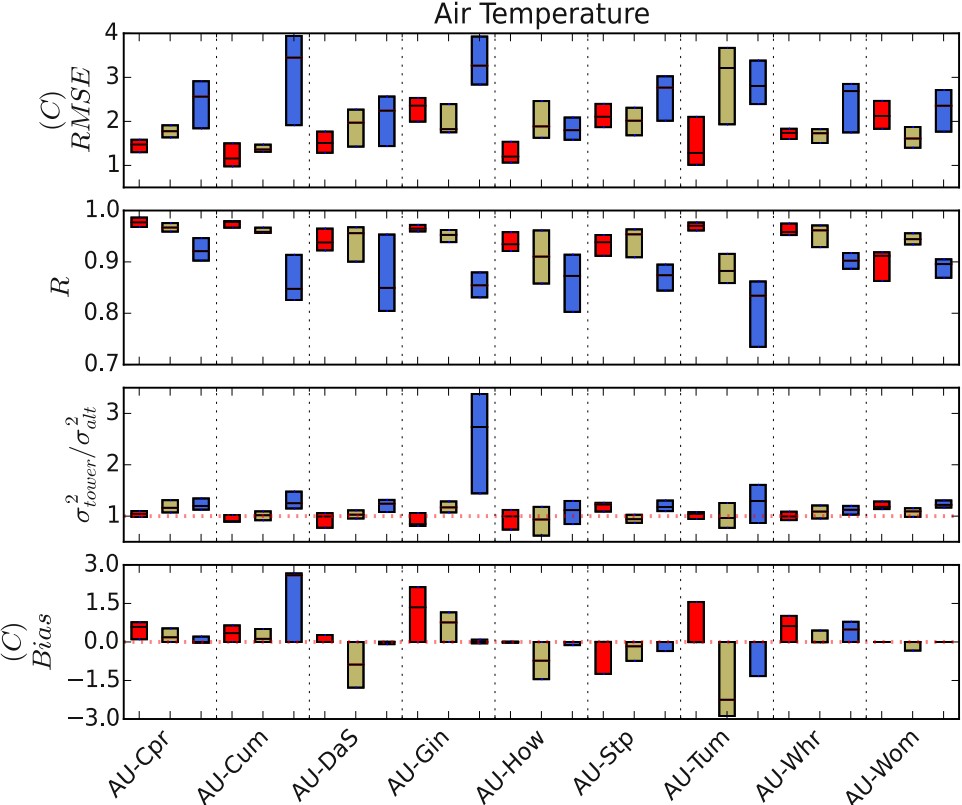

**Figure 5: Box plot of bias, variance ratio, correlation coefficient and root mean square error for the comparison of tower and alternate air temperature at 9 OzFlux sites. Statistics for AWS are plotted in red, ACCESS-R in green and ERA Interim in blue. Box lengths represent the 75% and 25% quartiles and the median is plotted as a solid line across the boxes.**



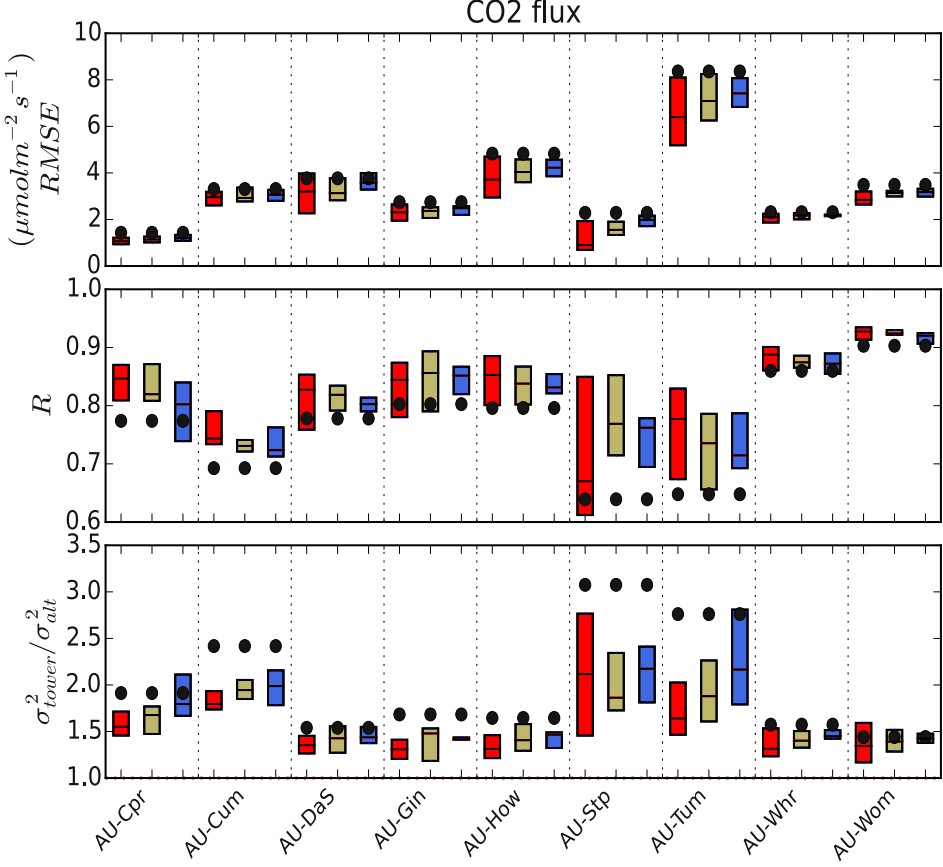

**Figure 6: Statistics for different gap filling window lengths for $CO_2$ flux. Window lengths are 60 days (red), 182 days (green), 365 (blue) and the whole length of the data set (black dots). Box lengths represent the 75% and 25% quartiles with the median plotted across the boxes as a solid line.**





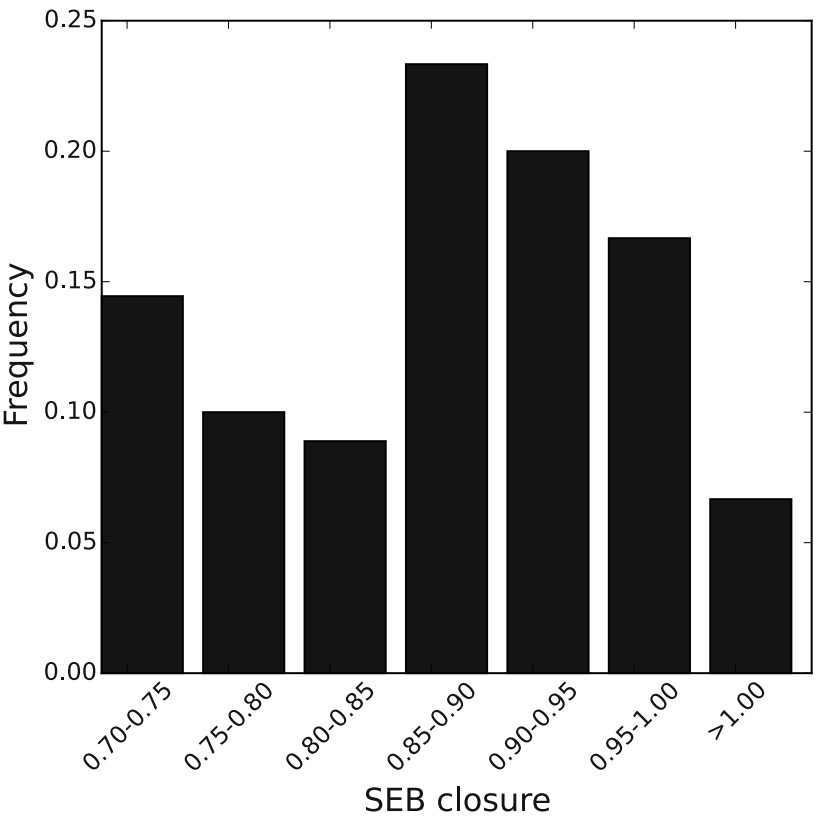

**Figure 7: Histogram of surface energy balance closure for 86 site-years of data across 23 sites.**




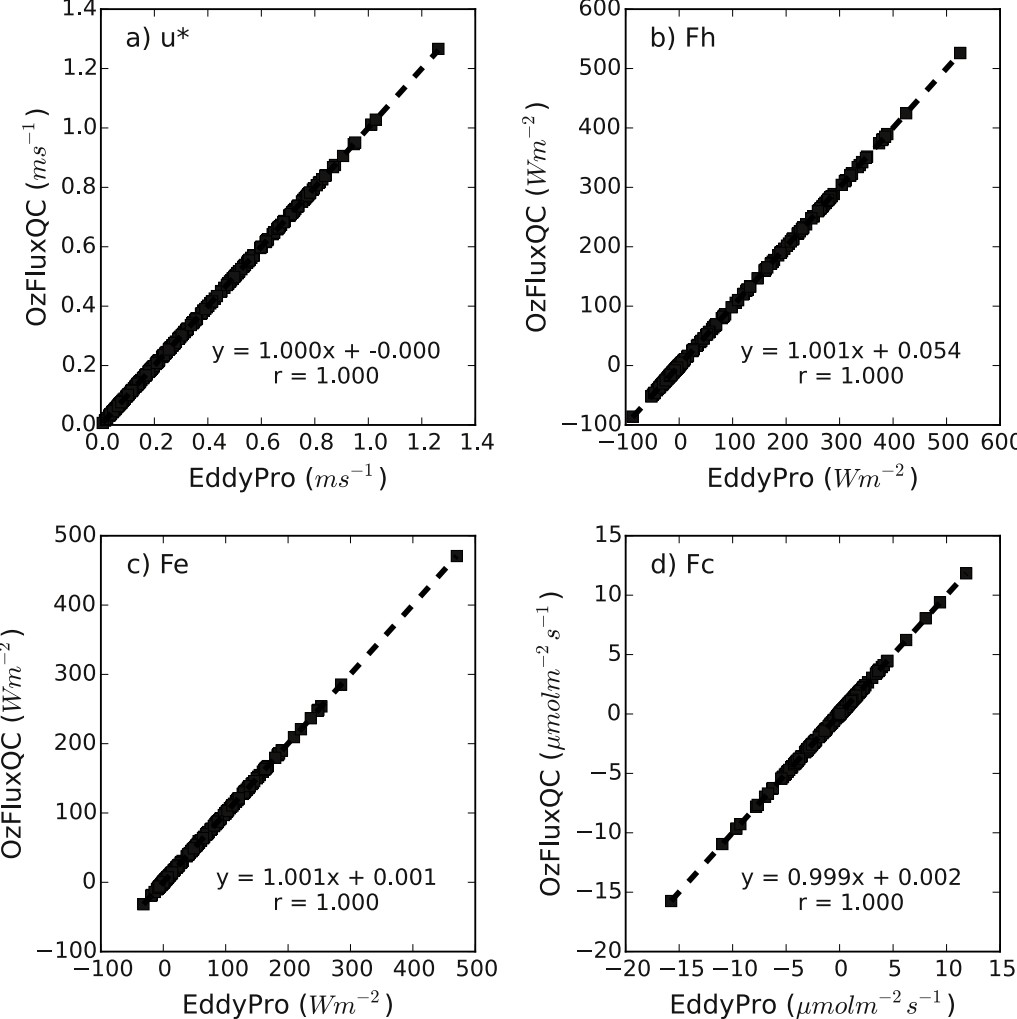

**Figure 8: Comparison of friction velocity (u*), sensible heat flux ($F_h$), latent heat flux ($F_e$) and CO$_2$ flux ($F_c$) calculated using EddyPro and OzFluxQC. Data are from the AU-How site for the period 2011/06/09 to 2011/08/05.**





**Figure 9: Bar chart of the annual sum of *NEE*, *GPP* and *ER* for 9 OzFlux sites in 2013. The bars, from left to right for each site, give the results from the nocturnal method with the Lloyd-Taylor respiration model (OzFluxQC, light red; REddyProc, mid red; FluxNet, dark red), the nocturnal method with an ANN respiration model (OzFluxQC, light green; DINGO, dark green) and the daytime method (OzFluxQC, light blue; FluxNet, dark blue).**