# Peer review of "OzFlux Data: Network integration from collection to curation"

_Biogeosciences, 2016_

## Referee Comment (RC1) · Anonymous Referee #1 · 23 Jun 2016

The paper provides a thorough description of the OzFlux data processing pipeline and the OzFluxQC software toolkit, without being overly detailed. It also presents interesting comparisons of Net Ecosystem Exchange and derived data products such as Ecosystem Respiration, highlighting effects from using different methods from the literature to create these products. It is clear that the task of creating such comparisons is considerably simplified by the availability of uniform data formats and tools such as the ones described in the paper. Furthermore, the information in this paper provides invaluable insight for this Special Edition of Biogeosciences on OzFlux. On the negative side, some of the analyses and results drawn in the paper could be seen as not being fully supported by the evidence presented, in particular the comparison of data products. It seems likely that additional information has to be included to properly support conclusions or at least more detailed clarification on limitations and assumptions

should be added. From the presentation point of view, the paper is very well written and properly structured, requiring only minor corrections.

Besides the detailed explanation of methods used in the OzFlux data processing pipeline, one of the most interesting contributions of the paper is the comparison of the effects of selecting different methods to generate a data product. This is a very relevant problem to the eddy covariance community, especially when it comes to improving repeatability of experiments. With that in mind, a question that naturally comes to mind is: how can the authors guarantee that the differences observed are due to differences in the methods and not in the implementation/coding of these methods? Many attempts to replicate experimental results from a paper using a re-implementation of a data processing method will lead to very different results. In many cases, this happens for reasons ranging from differences in the selection of parameter that was deemed minor (like a window size for gap filling) to misinterpretation of non-algorithmic portions of methods (like thresholds or exception cases). One of the ways to assess these differences can be to compare the results from a new implementation against results for the same input data using the original code. For some of the methods, the authors indicate they had access to the original code (e.g., the SOLO ANN method), but there is no indication of cross-validation of results. What about the other methods? This is not to say the differences pointed out by the authors are not due to differences in the methods – this actually seems the most likely explanation to me – but it's not clear the possibility of implementation differences was ruled out.

In a somewhat related note, how "the redundant steps at L3" are skipped when already applied is not very clear. Is this done from metadata generated by other software packages, does OzFluxQC try to determine that from the data, or is this dependent on user input? This is particularly relevant in the comparison with EddyPro results. How can the results really be equivalent if steps such as filtering based on stationarity checks (mentioned in the paper) are not applied? At the same time, comparing results from EddyPro using processing options selected to match what is being done by OzFluxQC

(when getting datalogger data directly) does not seem like a fair comparison. All the disabled options that would ideally be applied for a site's dataset can have significant impact on the computation of fluxes. It seems to me that for this comparison to be complete (and for the conclusion that both options can indeed be used interchangeably), many more checks would have to be done. This should cover looking at effects of enabling recommended corrections in EddyPro that are not available in OzFluxQC, and quantifying differences. It would also potentially involve looking at a range of sites and site conditions and the corresponding effects. That said, to me this should not be the focus of the paper. So my suggestion is to remove this comparison (leaving the detailed version for a follow-up paper), and focus on exploring the NEE/RE/GPP comparisons in more detail as suggested above. This would entail removing the paragraphs that make up section 5.2 and Figure 8, which would not impact the paper significantly, in my opinion.

The following comments are related to other minor issues.

It is not clear in the paper how the alternative data from AWS, ACCESS-R and ERA-Interim are reconciled and used in the gap filling process. How one (or more?) of the three sources are selected and used for gap filling of driver variables?

Also on gap filling, at the end of section 3.5, the issue of gaps being close to the window size can seriously restrict the applicability of gap filling methods. A bit more detail on the "minimum points criteria" mentioned in the paper would be very helpful in clarifying how this problem is addressed.

At the end of section 3.6, there is a description of final NEE, ER, and GPP time series, from what seems to be a combination of the results or the three used for partitioning. However, the same last few lines seem to indicate that NEE and GPP obtained from the hyperbolic light response curve method are not included in this these final versions of the variable. Is this correct? If not, how is the light response curve results incorporated? If yes, it's not clear why this method would be included in OzFluxQC (and in the paper).

The sentence below (from page 5, lines 23-24) is a bit awkward and could be rephrased for clarity: "The large range in precipitation amount and seasonality has resulted in a large range of biomes across Australia and OzFlux samples the majority"

Regarding the use of the CF Conventions, how are variables not covered in the current version of the conventions handled? This is probably worth one or two sentences for clarification.

For all gap filling comparison plots, it would be relevant to include indications of the amounts of missing data. These can be good indicators of variability in the data (or lack of variability, for some gap filling methods).

Also on the plots that use color, adding a color legend would make the plots considerably more readable compared to the text descriptions of the color associations in the figure captions.

On page 14, line 27: "drives" should be "drivers"

Section 4, which describes the data portal and data distribution, somewhat breaks the flow of the paper. Maybe moving the text in this section to section 2 would make the discussion on the data pipeline more fluid.

Finally, there is an issue that is not necessarily a discussion for this paper, but very relevant. There is a natural conflict between trying to encourage site teams to "know thy site" while providing the comfort of using standard (and vetted) software toolkit. In many cases, even experienced eddy covariance researchers will be tempted to not worry or question much about the results obtained using highly automated tools. Offering visual presentation of intermediate results, as done by OzFluxQC, is certainly a step in the right direction and is appreciated.

---

## Referee Comment (RC2) · Anonymous Referee #2 · 8 Jul 2016

The authors present a clear description of the data and processing from the OzFlux eddy-covariance network, which should serve as a go-to for future papers that make use of OzFlux data. I include minor comments below that I hope will help the authors to improve the manuscript. My only major comment is perhaps beyond the scope of the current dataset version, but I would like to emphasize it regardless in the hope that the authors will prioritize it in future efforts. Uncertainty. A variety of approaches exist for the estimation of uncertainty due to random noise and uStar thresholds, but no quantification of uncertainty is included in the current OzFlux data release. Given the importance of knowing what confidence to place in a particular observation I find this very disappointing. I hope the authors will continue their excellent work in the future by including uncertainty estimates for their data.

Abstract, line 23-26: These lines could be removed from the abstract as telling the

reader that processing used python, netCDF, OPeNDAP, etc is not really necessary at this stage.

Page 2, line 18: Ameriflux was officially formed in 1996, though initial papers did not come out until 1999. I am not sure the Pryor et al. 1999 paper is the most appropriate, as it is just the first paper to come out that uses ameriflux data and has not been highly cited. Other initial papers from Ameriflux that had a larger impact were, for example:

Hollinger et al. 1999: Seasonal patterns and environmental control of carbon dioxide and water vapour exchange in an ecotonal boreal forest, GLOBAL CHANGE BIOLOGY, 5, 891-902 Wilson and Baldocchi, 2000: Seasonal and interannual variability of energy fluxes over a broadleaved temperate deciduous forest in North America, AGRICULTURAL AND FOREST METEOROLOGY, 100, 1-18 Schmid et al. 2000: Measurements of CO2 and energy fluxes over a mixed hardwood forest in the mid-western United States AGRICULTURAL AND FOREST METEOROLOGY 103, 357-374

Page 2, line 20: It might be worth mentioning the FLUXNET network specifically here, and citing Baldocchi et al., 2001 FLUXNET: A new tool to study the temporal and spatial variability of ecosystem-scale carbon dioxide, water vapor, and energy flux densities, BULLETIN OF THE AMERICAN METEOROLOGICAL SOCIETY, 82, 2415-2434

Page 6, line 22: are also collected at all sites?

Page 8, line 26: It is not clear why three different sources of alternative met data are required. Does one provide information the others do not? An opening sentence justifying the need to have different sources of met data would help the reader.

Page 10, line 92: wide spread should be one word

Page 11, Section 3.2 The difference between OzFluxQC and DINGO could be better articulated here. The text only highlights some differences in the plots generated and the format of output (csv vs netCDF) but these differences are only superficial. What are the differences in terms of the internal processing? Do both approaches use the

same algorithms for gap-filling and partitioning, and if not, what are the fundamental differences?

Page 13, Line 10 It appears that the uStar threshold identified can vary seasonally and from year to year. As the uStar threshold can have a large impact on the fluxes (particularly flux partitioning) it would seem important to highlight better whether uStar varied seasonally or not, and at what sites. Have the authors considered using also a fixed uStar threshold? This is also included for comparison in the Fluxnet 2015 data release.

Page 14, line 5: Usually -> often

Page 21, line 10: The optimal window size likely changes depending on the time of year and the site. The authors claim that they have found an optimal size of 60 days seems somewhat ad-hoc. No evidence is presented nor any methodology for determining the optimal window size given. Perhaps rephrase. A more complex ANN design might not be necessary. Simply including day of year as one of the predictors should allow flexibility in the response.

Page 21, line 19: As there are no measurements of ER, you cannot really claim that daytime partitioning methods overestimate ER. The true value is unknown.

---

## Author Response (AR1)

**Response to reviewer comments**

**bg-2016-189: OzFlux Data: Network integration from collection to curation; Isaac et. al.**

We would like to thank the 2 reviewers for the time they have taken to read the manuscript and for their comments, suggested changes and proposed additions.

We begin with the comments from Reviewer 1 and for each reviewer we deal with their general comments first and then their specific points. Original wording from the reviewers comments are given in plain text and our responses are given in *italics*.

**Reviewer 1**

**General comments**

**Paragraph 1**

"On the negative side, some of the analyses and results drawn in the paper could be seen as not being fully supported by the evidence presented, in particular the comparison of data products. It seems likely that additional information has to be included to properly support conclusions or at least more detailed clarification on limitations and assumptions should be added."

*We believe that the reviewer's concerns expressed here will be addressed by the changes suggested under Paragraph 3 below.*
***DONE***

**Paragraph 2**

"How can the authors guarantee that the differences observed are due to differences in the methods and not in the implementation/coding of these methods?"
"For some of the methods, the authors indicate they had access to the original code (e.g., the SOLO ANN method), but there is no indication of cross-validation of results."

*The primary purpose of the current paper is to document the processing techniques underlying the data used for this special issue and the data available to researchers from OzFlux and from the FluxNet 2015 synthesis. While cross-validation of different implementations of the same algorithm is an important task (the numerical modelling community has gone before us here), it needs to be done very carefully (see Fratini and Mauder, 2014) and in our opinion is better suited to a stand alone paper. A full investigation would need to cover not just different implementations of, say, the same u\*-threshold detection technique, the same respiration model etc but also the gap filling techniques (marginal distribution sampling, ANN etc) used for the meteorological drivers.*
*To the specific point, the SOLO ANN, mentioned by the reviewer, the original code is called directly from OzFluxQC, we did not implement this in a different language. By definition, cross-validation of this would show identical results.*
*The current paper does introduce the issue, arguably to a limited extent due to the priority given to other material, by presenting results from different implementations of the same method in Figure 9 (OzFluxQC, ReddyProc and FluxNet nocturnal methods using the Lloyd-Taylor ER model, OzFluxQC and FluxNet day time methods using the Lasslop method) and it is discussed in the second paragraph of Section 7 Future Directions as an important area for further work.*

**NOTED BUT NOTHING DONE**

**Paragraph 3**

"How "the redundant steps at L3" are skipped when already applied is not very clear. Is this done from metadata generated by other software packages, does OzFluxQC try to determine that from the data, or is this dependent on user input?"

*We will amend the last sentence of Section 3.1 to clarify that the skipping of redundant steps requires user input but that OzFluxQC will fail with an error message if the choice made by the user is not consistent with the input data.*
**DONE**

"This is particularly relevant in the comparison with EddyPro results. How can the results really be equivalent if steps such as filtering based on stationarity checks (mentioned in the paper) are not applied? At the same time, comparing results from EddyPro using processing options selected to match what is being done by OzFluxQC (when getting datalogger data directly) does not seem like a fair comparison."
"So my suggestion is to remove this comparison (leaving the detailed version for a follow-up paper), and focus on exploring the NEE/RE/GPP comparisons in more detail as suggested above. This would entail removing the paragraphs that make up section 5.2 and Figure 8, which would not impact the paper significantly, in my opinion."

*The reviewer suggests removing Section 5.2 Comparison with EddyPro and the associated Figure 8 and to "... focus on exploring the NEE/ER/GPP comparisons in more detail as suggested above."*
*We are happy to remove Section 5.2 and Figure 8 and to leave a more detailed comparison of OzFluxQC and other flux processing tools (such as EddyPro) to a subsequent paper. This is consistent with the primary focus of the paper as set out in our response to Paragraph 2 of the reviewer comments above.*
*We are also happy to further explore the comparison of NEE, ER and GPP but we are not completely sure what the reviewer means by "... as suggested above" and would appreciate their comments on the following proposal.*
*The existing Figure 9 shows annual sums of NEE, ER and GPP for 9 OzFlux sites in 2013 calculated using 7 different methods, 3 of which are implemented in OzFluxQC, 2 of which come from the FluxNet 2015 synthesis data set and 1 each from ReddyProc and DINGO. The figure shows that while the 7 different values for NEE, ER and GPP show the same site to site variability, there is considerable variability between methods within individual sites. The variability between methods at individual sites is discussed in the existing manuscript but not pursued.*
*We propose to expand the comparison of NEE, ER and GPP presented in the existing Figure 9 by focussing on annual sums of NEE, ER and GPP for 3 sites, AU-Cpr, AU-DaS and AU-Whr, from 2 groups of methods; nocturnal, u\*-filtered (OzFluxQC, FluxNet and ReddyProc) and light response curve (LRC) intercept (OzFluxQC and FluxNet). Our objective is to explore the variability between the nocturnal methods and between the LRC methods as functions of u\* threshold (nocturnal only) from the 3 implementations, of gap filling techniques (all methods) and of parameter values from the non-linear curve fit process (all methods). The hypotheses are:*
   a) *that variability between methods at the sites tested can be primarily attributed to one of the possible causes listed above and*
   b) *that this information can be used to direct efforts aimed at resolving discrepancies between different implementations of the same method.*
*Most of the additional analysis required by this proposal can be done with data prepared for the original manuscript. The new material would be presented as Section 5.3 Comparison of Methods (the existing Section 5.3 will be renamed to Section 5.2 following the removal of the existing Section*

*5.2).*

*DONE (SEE BELOW)*

*We have had to approach this aspect of the reviewers comments in a slightly different manner than that presented above due to limitations in the intermediate data saved from the original analysis. As an alternative, we have presented time series plots of monthly sums of NEE, GPP and ER for the 3 sites showing the range in values from all 7 methods available (see section 4.2 for a description of these) and comparing these to the range in values due to the uncertainty in the u\* threshold value from a single method (nocturnal, u\*-filtered, ER from Lloyd-Taylor as implemented in OzFluxQC). The results are presented in a new figure (Figure 9) and discussed in a new section (Section 4.3).*

*We believe that the added figure and text achieve the same end as the original proposal and highlight the relative contributions to uncertainty arising from (a) different implementations of the same algorithm in different packages, (b) different methods available within a single package and (c) the effect of uncertainty in the u\* threshold value.*

*In addition, the Associate Editor raised a point regarding the proposed sites for this analysis. We have used the proposed sites and address the Associate Editor's point in a separate response at the end of this document.*

**Specific comments**

1) It is not clear in the paper how the alternative data from AWS, ACCESS-R and ERA-Interim are reconciled and used in the gap filling process. How one (or more?) of the three sources are selected and used for gap filling of driver variables?

   *This is a good point. To address this, we propose to add a short paragraph (3 sentences should suffice) between the existing paragraphs 1 and 2 on page 14 of the current manuscript. The paragraph will emphasise the differences in data available from AWS, ACCESS-R and ERA_Interim (these are introduced in Sections 2.5.1, 2.5.2 and 2.5.3), describe the default hierarchy of choice made by OzFluxQC, how this can be changed by the user, the circumstances which may justify such a change and the evidence provided to support such a change.*
   *DONE*

2) Also on gap filling, at the end of section 3.5, the issue of gaps being close to the window size can seriously restrict the applicability of gap filling methods. A bit more detail on the "minimum points criteria" mentioned in the paper would be very helpful in clarifying how this problem is addressed.

   *We will address this point by re-writing the 2 sentences at the end of Section 3.5 to clarify the automated method used by OzFluxQC to determine the optimal window size when the gap size approaches or exceeds the default window size. The description of the role of the minimum points criteria will also be clarified.*
   *DONE*

3) At the end of section 3.6, there is a description of final NEE, ER, and GPP time series, from what seems to be a combination of the results or the three used for partitioning. However, the same last few lines seem to indicate that NEE and GPP obtained from the hyperbolic light response curve method are not included in this these final versions of the variable. Is this correct? If not, how is the light response curve results incorporated? If yes, it's not clear why this method would be included in OzFluxQC (and in the paper).

   *The reviewer is correct. The 3 methods for estimating ER, and hence gap filling NEE and*

*calculating GPP, are introduced at the start of Section 3.6 but it is not made clear that all 3 ER estimates and the corresponding gap-filled NEE and GPP, are available as separate outputs from OzFluxQC. We will address this point by adding a sentence to the end of Section 3.6 that makes it clear values of ER, NEE and GPP from all 3 methods are output by OzFluxQC and that the user is free to choose which one they believe is the best for their site. We will also introduce subscript notation in Eqn. 1 and 2 and in the use of ER in the third paragraph of Section 3.6 to clearly indicate the 3 sources of ER estimates. Eqn. 4 will be expanded to contain 3 equations for GPP with subscripts indicating the different sources of ER.*
***DONE***

4) The sentence below (from page 5, lines 23-24) is a bit awkward and could be rephrased for clarity: "The large range in precipitation amount and seasonality has resulted in a large range of biomes across Australia and OzFlux samples the majority"

   *The wording of the sentence will be changed to clarify the sentence meaning.*
   ***DONE***

5) Regarding the use of the CF Conventions, how are variables not covered in the current version of the conventions handled? This is probably worth one or two sentences for clarification.

   *The reviewer makes another good point. A sentence describing the treatment of variables not covered by the CF Conventions will be added to the preamble of Section 3. A further sentence will be added to Section 7 Future Directions that notes our intention to propose to the CF Working Group standard names for those flux data set variables that are not currently covered by the CF Conventions.*
   ***DONE***

6) For all gap filling comparison plots, it would be relevant to include indications of the amounts of missing data. These can be good indicators of variability in the data (or lack of variability, for some gap filling methods).

   *The percentage of missing data for each site will be added as text above the top axis of the top panel in each of Figures 4, 5 and 6.*
   ***DONE***

7) Also on the plots that use color, adding a color legend would make the plots considerably more readable compared to the text descriptions of the color associations in the figure captions.

   *Figures 4, 5, 6 and 9 will be changed to include a colour bar as suggested.*
   ***DONE***

8) On page 14, line 27: "drives" should be "drivers"

   *The wording will be changed as suggested.*
   ***DONE***

9) Section 4, which describes the data portal and data distribution, somewhat breaks the flow of the paper. Maybe moving the text in this section to section 2 would make the discussion on the data pipeline more fluid.

*This is a good suggestion and we will move Section 4 to become Section 2.6. Our intention with the original structure was reflect the typical work flow of a site PI (one processes data and then makes it available) but as the reviewer points out, this is not necessarily the best structure for a research article.*
***DONE***

10) Finally, there is an issue that is not necessarily a discussion for this paper, but very relevant. There is a natural conflict between trying to encourage site teams to "know thy site" while providing the comfort of using standard (and vetted) software toolkit. In many cases, even experienced eddy covariance researchers will be tempted to not worry or question much about the results obtained using highly automated tools. Offering visual presentation of intermediate results, as done by OzFluxQC, is certainly a step in the right direction and is appreciated.

*This is an excellent comment by the reviewer and goes straight to the heart of a debate that has raged within the first author for years. On the one hand, we want to minimise the time researchers spend in the drudgery of data processing so they have more time for the interpretation of results. On the other hand, sometimes one simply has to look at data, and at lots of data, in order to obtain a deeper understanding what an ecosystem is doing. The pressure to produce quick results does not help this process of deep involvement in the data. The first author has not found a foolproof way of helping others to walk this fine line and so feels unqualified at this stage to put anything into writing. The most effective way has been practical demonstrations of the need to visually examine and to rigorously question the data and the best time to do this demonstration has been during training exercises in the use of the software.*
*We acknowledge the reviewer's insight but feel the topic is worthy of it's own paper and that to insert something here could not do the subject justice.*
***NOTED BUT NOTHING DONE***

**Reviewer 2**

**General comments**

**Paragraph 1**

"My only major comment is perhaps beyond the scope of the current dataset version, but I would like to emphasize it regardless in the hope that the authors will prioritize it in future efforts. Uncertainty."

*The reviewer makes a very important point. The natural extension of the work presented here, and a natural extension to the OzFluxQC software, is the estimation of uncertainty in NEE, ER and GPP values. As the reviewer hints, we feel that this is beyond the scope of the present work. Estimation of uncertainty is discussed in Section 7 Future Directions as the third topic (fourth paragraph) after harmonising OzFlux data with FluxNet and resolving disparities in implementations of the same partitioning algorithms.*
*The estimation of uncertainty in this dataset will be the subject of future work across OzFlux. We have already developed code to estimate uncertainty due to random error, model error and u\*-threshold uncertainty and this has been tested at a single site in the last week. However, there is still significant work to be done to fully characterise the uncertainty such as resolving the differences between partitioning results from OzFluxQC, ReddyProc and FluxNet, estimating the effect of different gap-filling methods and different gap fractions and assessing the uncertainty and bias due to the lack of profile measurements of $CO_2$ concentration within canopies (only 5 out of*

*the 15 sites with substantial canopies currently have profile systems). We feel it is better to treat this topic comprehensively in a separate paper.*
**NOTED BUT NOTHING DONE**

**Specific comments**

1) Abstract, line 23-26: These lines could be removed from the abstract as telling the reader that processing used python, netCDF, OPeNDAP, etc is not really necessary at this stage.

   *This is a good point that will result in a shorter and more concise Abstract. We will remove the sentences referred to in the above comment.*
   **DONE**

2) Page 2, line 18: Ameriflux was officially formed in 1996, though initial papers did not come out until 1999. I am not sure the Pryor et al. 1999 paper is the most appropriate, as it is just the first paper to come out that uses ameriflux data and has not been highly cited. Other initial papers from Ameriflux that had a larger impact were, for example:
   1) Hollinger et al. 1999: Seasonal patterns and environmental control of carbon dioxide and water vapour exchange in an ecotonal boreal forest, GLOBAL CHANGE BIOLOGY, 5, 891-902
   2) Wilson and Baldocchi, 2000: Seasonal and interannual variability of energy fluxes over a broadleaved temperate deciduous forest in North America, AGRICULTURAL AND FOREST METEOROLOGY, 100, 1-18
   3) Schmid et al. 2000: Measurements of CO2 and energy fluxes over a mixed hardwood forest in the mid-western United States AGRICULTURAL AND FOREST METEOROLOGY 103, 357-374

   *Many thanks to the reviewer for correcting the historical inaccuracies in our Introduction. We will amend the text as suggested.*
   **DONE**

3) Page 2, line 20: It might be worth mentioning the FLUXNET network specifically here, and citing:
   1) Baldocchi et al., 2001 FLUXNET: A new tool to study the temporal and spatial variability of ecosystem-scale carbon dioxide, water vapor, and energy flux densities, BULLETIN OF THE AMERICAN METEOROLOGICAL SOCIETY, 82, 2415-2434

   *This is also a good point and would also serve to introduce the FluxNet syntheses mentioned on line 31 of the same page. We will amend the text as required.*
   **DONE**

4) Page 6, line 22: are also collected at all sites?

   *This is correct, the additional quantities mentioned are collected at all OzFlux sites. We will amend the text as suggested.*
   **DONE**

5) Page 8, line 26: It is not clear why three different sources of alternative met data are required. Does one provide information the others do not? An opening sentence justifying the need to have different sources of met data would help the reader.

   *The reviewer makes a valid point. Although the different quantities supplied by the 3*

*sources of alternate data are given in Sections 2.5.1, 2.5.2 and 2.5.3, it is somewhat buried among other detail.   We will insert a sentence in Section 2.5 Ancillary Data to better explain the need for the 3 sources of alternate data.*
***DONE***

6) Page 10, line 92: wide spread should be one word

*The text will be amended as suggested.*
***DONE***

7) Page 11, Section 3.2 The difference between OzFluxQC and DINGO could be better articulated here. The text only highlights some differences in the plots generated and the format of output (csv vs netCDF) but these differences are only superficial. What are the differences in terms of the internal processing? Do both approaches use the same algorithms for gap-filling and partitioning, and if not, what are the fundamental differences?

*The Dynamic Integrated Gap-filling and partitioning tool for OzFlux (DINGO) is described in Beringer et al (2016), a paper in the same Special Issue as the current paper.  As far as possible, we would like to avoid duplication of material between these 2 papers but we also acknowledge the frustration of having to flick between the 2 papers to compare methods. We will re-write the second paragraph of Section 3.2 The Roles of OzFluxQC and DINGO to briefly describe the main similarities and differences in the methods and algorithms used* ***DONE***.

8) Page 13, Line 10 It appears that the uStar threshold identified can vary seasonally and from year to year. As the uStar threshold can have a large impact on the fluxes (particularly flux partitioning) it would seem important to highlight better whether uStar varied seasonally or not, and at what sites. Have the authors considered using also a fixed uStar threshold? This is also included for comparison in the Fluxnet 2015 data release.

*On re-reading Section 3.4, we feel there are 2 parts to this comment.  The first is that the existing text does not include a description of how OzFluxQC uses the u\* threshold information from the CPD method.  We will address this by splitting the second paragraph at the sentence beginning "At some OzFlux sites, ..." (page 13, line 27) and adding a sentence to the now truncated second paragraph that describes how OzFluxQC implements the u\* thresholds found by the CPD method.  This will include the information that, by default, OzFluxQC uses annual values of the u\* threshold but that the user can also specify the threshold to be used as a function of time.*
***DONE: Second paragraph split, truncated second paragraph appended to first paragraph, new second paragraph added to explain use of u\* threshold by OzFluxQC.***
*The second part, and this may have been the reviewers main point, concerns seasonal variability in the u\* threshold values detected by the CPD method.  Our implementation of the Barr et al (2013) CPD technique provides the u\* threshold for each 1000 point "season" but does not record the start and end dates for the season making it hard to relate the 1000 point "seasons" to any seasonal changes in ecosystem phenology.*
*As a compromise, we propose adding text and a table to Section 3.4 to describe the intra-annual and inter-annual variability in the u\* threshold values detected by the CPD method at the 9 OzFlux sites used in Figures 4, 5, 6 and 9 and testing any intra-annual variation for significance.*
***DONE: analysis of the u\* threshold results for the 9 sites presented in the paper showed no significant seasonal variability.  This is likely to be due to the fact that there is no significant change in the structure of the vegetation canopy, and hence roughness length,***

*with season in the ecosystems studied. All are evergreen, native ecosystems.*
*Rather than insert a table containing a null result, we have chosen to include a paragraph*
*in Section 4.3 Comparison of Methods and Uncertainty that discusses the lack of seasonal*
*variability in the u\* threshold at the 9 sites studied.*

9) Page 14, line 5: Usually -> often

*The text will be amended as suggested.*
*DONE*

10) Page 21, line 10: The optimal window size likely changes depending on the time of year and the site. The authors claim that they have found an optimal size of 60 days seems somewhat ad-hoc. No evidence is presented nor any methodology for determining the optimal window size given. Perhaps rephrase. A more complex ANN design might not be necessary. Simply including day of year as one of the predictors should allow flexibility in the response.

*In the existing text, we state that the "best performance" is achieved with a window size of*
*60 days and that "performance degrades with increasing window length.". We did not*
*intend to imply that 60 days was the "optimal window size" and accept that the wording in*
*paragraph 5 of Section 6 Conclusions is ambiguous. We will amend the text along the lines*
*suggested by the reviewer to clarify this point.*
*DONE*

11) Page 21, line 19: As there are no measurements of ER, you cannot really claim that daytime partitioning methods overestimate ER. The true value is unknown.

*The reviewer is correct, the offending wording is sloppy. The sentence will be re-written to*
*emphasise that the under-estimation is relative to the other methods.*
*DONE*

**Response to Associate Editor Comments**

The Associate Editor has made several points in addition to the reviewer's comments and we address those points here.

1) Discussion of the importance of uncertainty quantification.
   *We have added a short paragraph at the end of Section 3.6 that discusses the possible sources of uncertainty in NEE, GPP and ER and their importance.*

2) Re-implementation of codes and use of different data sets leading to errors.
   *We have added a short paragraph at the end of Section 3.1 that discusses the possibility of errors caused by re-implementation of the same algorithm in different languages.*

3) Use of AU-Tum data.
   *We have elected to continue with the 3 sites originally proposed (AU-Cpr, AU-DaS and AU-Whr) for the following reasons.*

   *The results for AU-Tum from the 7 different methods available shows the largest variation but this is an "artefact" caused by differences in the data sets used by the different analyses, differences in the analysis methods and the presence of strong advection at this site due to nocturnal drainage flows. Of course, in reality it is precisely such "artefacts" that are of interest in detailed studies for they show where general methods break down and in quantifying this help to place confidence intervals on numbers from less complex sites with more consistent data sets.*

   *We feel that while this is an interesting and useful point, a full explanation as to why this site shows the largest variability would require too much detail for what is intended to be a general paper describing methods adopted across the OzFlux network. Central to the philosophy behind the OzFlux data processing path is the idea of having a "default" data set for each site, obtained using default methods and settings, and a "site_pi" data set provided by the site PI themselves using what they believe to be the best methods and settings for their site. In the case of complex sites such as Tumbarumba, OzFlux would always recommend the site PIs data set over the default. The central Australian sites of Alice Springs Mulga and Ti Tree East are in the same category.*

[revised manuscript text omitted]